

# Living on the walls of super-QCD

**Vladimir Bashmakov** [1], **Francesco Benini** [2,3⋆],
**Sergio Benvenuti** [2] **and Matteo Bertolini** [2,3]

**1** Dipartimento di Fisica, Università di Milano-Bicocca
and INFN, Sezione di Milano-Bicocca, I 20126 Milano, Italy
**2** SISSA and INFN – Via Bonomea 265; I 34136 Trieste, Italy
**3** ICTP – Strada Costiera 11; I 34014 Trieste, Italy

⋆ fbenini@sissa.it

## Abstract

We study BPS domain walls in four-dimensional $\mathcal{N} = 1$ massive SQCD with gauge group $SU(N)$ and $F < N$ flavors. We propose a class of three-dimensional Chern-Simons-matter theories to describe the effective dynamics on the walls. Our proposal passes several checks, including the exact matching between its vacua and the solutions to the four-dimensional BPS domain wall equations, that we solve in the small mass regime. As the flavor mass is varied, domain walls undergo a second-order phase transition, where multiple vacua coalesce into a single one. For special values of the parameters, the phase transition exhibits supersymmetry enhancement. Our proposal includes and extends previous results in the literature, providing a complete picture of BPS domain walls for $F < N$ massive SQCD. A similar picture holds also for SQCD with gauge group $Sp(N)$ and $F < N + 1$ flavors.

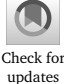

# 1  Introduction and summary of results

Interesting progress has been recently made in understanding the dynamics of quantum field theories (QFTs) in three space-time dimensions. This progress has also led to new insights (and surprises) on the relation between three-dimensional and four-dimensional QFTs. One concrete situation in which such a connection becomes manifest is when domain walls—which are codimension-one solitonic states that a QFT contains whenever there exist multiple vacua separated by a potential barrier—are present.

Notable examples of 4d theories with domain walls are Yang-Mills (YM) theory and QCD, at the special value $\theta = \pi$ of the topological theta term. In a nice series of papers [1–3] (see also [4]), a rather complete picture of the vacuum dynamics of YM and QCD and their domain walls has been proposed. While it is believed that $CP$ is an exact quantum symmetry when $\theta = 0$, the authors gave arguments supporting the claim that for $\theta = \pi$ $CP$ is spontaneously broken in two degenerate gapped vacua. Hence, domain walls exist connecting these two vacua.

The effective dynamics on YM domain walls is gapped and captured by a Chern-Simons (CS) topological quantum field theory (TQFT) [2]. On the contrary, QCD domain walls behave rather differently depending on the quark masses [3]. For large quark masses compared to the QCD scale $\Lambda_{\text{QCD}}$ their low energy dynamics is as in YM, while for small masses there are massless excitations on the domain walls—Goldstone bosons for broken symmetries—described by a non-linear sigma model (NLSM) with target $\mathbb{CP}^{F-1}$, where $F$ is the number of flavors. This implies that at some value $m_{\text{4d}}^*$ of the quark masses, a phase transition on the domain walls should occur. This picture has been later confirmed within a pure holographic context [5].

One of the key ideas in [2,3] is that one can capture all low-energy properties of domain walls by identifying their three-dimensional worldvolume theory, and studying its dynamics. This gives direct connections between phases of 4d theories, the domain walls they support, and recent advances in charting the phase diagram of 3d theories and their dual descriptions (see *e.g.* [6–19]).

In this paper we discuss another class of theories which admits a rich variety of domain walls, namely 4d $\mathcal{N} = 1$ massive super-QCD (SQCD). For generic values of the continuous parameters—flavor masses and $\theta$ angle—the theory develops multiple isolated supersymmetric gapped vacua, where the gaugino bilinear condenses and confinement occurs. The number of vacua equals the dual Coxeter number $h$ of the gauge group $G$. For any pair of vacua, one can construct field configurations in which the theory sits in two different vacua on the left and right half-spaces, respectively. In such configurations, a domain wall must necessarily separate the two spatial regions. This gives rise to the aforementioned rich variety of domain walls.

When the gauge group is simply connected, the degenerate vacua arise from the spontaneous breaking of a discrete R-symmetry: they arrange as the $h^{\text{th}}$ roots of unity, and are cyclically rotated into each other by the broken R-symmetry. This implies that the vacua are

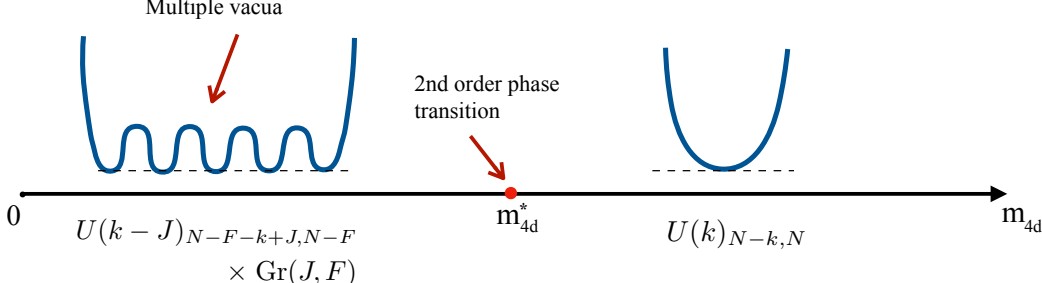

Figure 1: A qualitative picture of the phase diagram of $k$-walls of $SU(N)$ SQCD with $F < N$ flavors. In the small mass regime the $k$-wall theory has multiple vacua, parametrized by an integer $J$. In each vacuum, a topological theory is accompanied by a supersymmetric NLSM with target the complex Grassmannian $\mathrm{Gr}(J,F) = U(F)/\big[U(J) \times U(F-J)\big]$. A similar phase diagram holds for $Sp(N)$ SQCD with $F < N + 1$ flavors.

physically equivalent, and the properties of domain walls only depend on how many vacua we jump by. We call $k$-wall a domain wall connecting the $j^{\text{th}}$ vacuum to the $(j+k)^{\text{th}}$ vacuum. In $SU(N)$ SQCD, we have $0 < k < N$. Even for fixed topological sector $k$, there can be multiple physically-inequivalent degenerate domain walls that connect the very same two vacua. This of course is an effect of supersymmetry. Indeed, the 4d $\mathcal{N} = 1$ supersymmetry algebra admits a two-brane charge [20] and so the tension of domain walls enjoys a BPS bound. One can argue that SQCD walls saturate the bound—they are 1/2 BPS—so they preserves two supercharges, corresponding to $\mathcal{N} = 1$ supersymmetry in three dimensions.

In $SU(N)$ SQCD there is a qualitative difference between $F < N$ and $F \geq N$, where $F$ is the number of flavors [21–26]. The basic reason is that for $F \geq N$ also baryons, besides mesons, parametrize the moduli space. This suggests a somewhat different structure of the domain wall spectrum. In this paper we focus on the case $F < N$, leaving the case $F \geq N$ to future work [27].

The problem of understanding and classifying the BPS domain walls of SQCD is not new and there exists an extensive literature on the subject, which includes [28–48]. However, the improved understanding that we now have of the dynamics of $\mathcal{N} = 1$ three-dimensional CS-matter theories (see *e.g.* [12, 16–18, 49–52]), together with a few more facts which were not fully appreciated previously, let us reconsider this problem and provide a more complete and satisfactory picture, in the regime $F < N$. For instance, in Table 2 we list all BPS domain walls of $SU(N)$ SQCD for $N \leq 5$. Our findings include and extend on previous results, solving also a few puzzles that have been raised.

Our strategy is to provide a 3d worldvolume description of the low-energy effective dynamics on $k$-walls in 4d $\mathcal{N} = 1$ $SU(N)$ SQCD, with $F < N$ flavors, valid as the flavor mass $m_{4\text{d}}$ is varied, and capable of capturing 3d phase transitions. Our proposal is the three-dimensional $\mathcal{N} = 1$ CS theory

$$\text{3d } \mathcal{N} = 1 \quad U(k)_{N - \frac{k+F}{2}, N - \frac{F}{2}} \tag{1.1}$$

coupled to $F$ matter superfields $X$ transforming in the fundamental representation of $U(k)$. The theory has a real superpotential $\mathcal{W}(X, X^{\dagger})$, which includes a mass term $m \operatorname{Tr} X^{\dagger} X$ and two quartic terms.

The vacuum structure of the 3d theory (1.1) depends on the sign of $m$. For $m < 0$, that corresponds to small 4d mass $m_{4\text{d}}$ compared to the SQCD scale $\Lambda$, there are multiple vacua in which the low-energy effective theory is the product of a TQFT and a supersymmetric non-linear sigma model. Each vacuum corresponds to a different domain wall in the same soliton

sector. For $m > 0$, that corresponds to large $m_{4d}$, there is a single gapped vacuum hosting a TQFT. The effective theory in this vacuum agrees with the theory that Acharya and Vafa (AV) discovered to govern the dynamics on domain walls in $SU(N)$ SYM [42]. This is an important test of our proposal. At $m = 0$, corresponding to a 4d mass $m_{4d}^*$ whose precise value we cannot determine but that is of order $\Lambda$, there is a second-order phase transition separating the two phases, in which the multiple vacua of the $m < 0$ regime coalesce into one. Note that when such phase transition occurs, nothing special happens in the bulk—exactly the same phenomenon observed in [3] for QCD. The phase transition is described by a 3d $\mathcal{N} = 1$ SCFT. However, for the special values $F = 1$, $k = 1$ or $k = N - 1$, we conjecture that 3d supersymmetry is enhanced to $\mathcal{N} = 2$ at low energy on the domain wall. In the very special case of $SU(2)$ SQCD with 1 flavor, we conjecture that the SCFT on the 1-wall has enhanced $\mathcal{N} = 4$ supersymmetry. Figure 1 contains a qualitative picture of the low energy behavior of theory (1.1).

Our proposal passes several non-trivial checks. As already mentioned, in the limit $m_{4d} \gg \Lambda$ it reproduces the theory of Acharya-Vafa [42]. Moreover, since quartic interactions dominate over the mass term, the Witten index remains constant through the phase transition and equal to $\binom{N}{k}$. This is required from the 4d point of view because, as long as we keep the flavor mass positive, there cannot be leaking of states at infinity in field space. Note that constancy of the Witten index is realized in a rather non-trivial way: at small $m_{4d}$ one has to sum over the inequivalent degenerate walls.

Even more strikingly, in the small $m_{4d}$ regime we are able to explicitly construct the BPS domain walls as solitons of 4d SQCD in an almost semi-classical way. The new idea is to construct "hybrid" walls, combining standard domain walls of the Wess-Zumino type on the mesonic space with sharp transitions in an unbroken SYM sector that is present on the mesonic space. This construction *exactly* matches the intricate vacuum structure displayed by theory (1.1) at $m < 0$.

We repeat this whole analysis for 4d $\mathcal{N} = 1$ $Sp(N)$ SQCD with $F < N + 1$ flavors, finding a similar phase diagram. As a special case, we obtain the extension of the AV theory to symplectic groups: such a theory describes domain walls in $Sp(N)$ SYM.

The rest of the paper is organized as follows. In Section 2 we review some basic properties of domain walls in pure SYM. In Section 3 we recall the vacuum structure of $SU(N)$ SQCD for $F < N$, and summarize some properties that BPS domain walls should satisfy. Section 4 contains our proposal for the effective three-dimensional theory describing these domain walls and a thorough analysis of its vacuum structure. This analysis will already encode several non-trivial checks. In Section 5 we focus on the small 4d mass regime and explicitly construct, by a 4d analysis, the domain walls interpolating between SQCD vacua. The results we get exactly match the 3d analysis. Finally, in Section 6 we discuss domain walls in $Sp(N)$ massive SQCD.

## 2   Domain walls of SYM: a review

Let us consider four-dimensional super-Yang-Mills (SYM) theory with $\mathcal{N} = 1$ supersymmetry and gauge group $G$. For simplicity, we restrict to the case of simply-connected[1] gauge groups with simple algebra $\mathfrak{g}$. The classical $U(1)$ R-symmetry is anomalous, and in the quantum theory it gets reduced to a $\mathbb{Z}_{2h}$ subgroup, where $h = c_2(\mathfrak{g})$ is the dual Coxeter number of $\mathfrak{g}$.[2]

---

[1]When the group is the quotient of a simply-connected $G$ by a subgroup of its center, the number of vacua is the same as for $G$, but their physical properties are different [53]. In particular, the R-symmetry is a subgroup of the $\mathbb{Z}_{2h}$ discussed below, or completely absent.

[2]Recall that: $c_2(\mathfrak{su}(N)) = N$, $c_2(\mathfrak{so}(N)) = N - 2$ for $N \geq 5$, $c_2(\mathfrak{sp}(N)) = N + 1$, $c_2(\mathfrak{e}_6) = 12$, $c_2(\mathfrak{e}_7) = 18$, $c_2(\mathfrak{e}_8) = 30$, $c_2(\mathfrak{f}_4) = 9$, $c_2(\mathfrak{g}_2) = 4$.

The non-perturbative dynamics gives rise to a gaugino condensate that spontaneously breaks $\mathbb{Z}_{2h}$ to $\mathbb{Z}_2$ and provides $h$ gapped vacua rotated by the action of $\mathbb{Z}_h = \mathbb{Z}_{2h}/\mathbb{Z}_2$:

$$\langle \lambda\lambda \rangle = \Lambda^3 \, \omega^k \,, \tag{2.1}$$

where $\Lambda$ is the dynamically-generated scale, $\omega = e^{2\pi i/h}$ is the basic $h^{\text{th}}$ root of unity, and $k = 0, \ldots, h-1$ labels the vacua. In other words, in different vacua the gaugino condensate differs by a phase. We can describe the various condensates through an effective superpotential

$$W_{\text{SYM}} = h \left( \Lambda^{3h} \right)^{1/h} = h \, \Lambda^3 \omega^k \,. \tag{2.2}$$

This should be though of as the generating function of gaugino bilinears, in the sense that $\langle \lambda\lambda \rangle = \partial W_{\text{SYM}}/\partial \log \Lambda^{3h}$ [54].

Since the vacua ara gapped, there must exist domain walls—*i.e.* finite-tension codimension-one solitonic objects—connecting them. More precisely, one can consider phases in which in different spatial regions the theory sits in different vacua: those regions must be separated by dynamical domain walls. The 4d $\mathcal{N} = 1$ supersymmetry algebra admits a two-brane charge [20, 55], and as a consequence there can exist half-BPS saturated domain walls, whose tension is minimal within their soliton sector [28, 56]. Their "central charge" is twice the total excursion of the superpotential from one vacuum to the other [56, 57],

$$Z = 2\Delta W = 2e^{i\gamma}|\Delta W| \,, \tag{2.3}$$

with $\gamma$ the phase of $Z$, and the tension of a BPS domain wall is fixed by the supersymmetry algebra in terms of the superpotential as

$$T = |Z| = 2|\Delta W| \,. \tag{2.4}$$

For SYM, the tension of BPS walls connecting the $j^{\text{th}}$ vacuum to the $(j+k)^{\text{th}}$ vacuum is

$$T = 2\,|\Delta W_{\text{SYM}}| = 2h\,\Lambda^3 \left| \omega^k - 1 \right| \,. \tag{2.5}$$

This is an exact non-perturbative result.

Acting with the generator of the $\mathbb{Z}_{2h}$ R-symmetry, the phase of the gaugino is shifted by $e^{\pi i/h}$, and thus the phase of the gaugino condensate by the $h^{\text{th}}$ root of unity $e^{2\pi i/h}$.[3] Notice that, because of the anomaly of the continuous $U(1)_R$, R-symmetry rotations are accompanied by a shift of the theta angle from $\theta$ to $\theta + 2\pi$, which is a symmetry of the quantum theory. Employing R-symmetry rotations, we can restrict, without loss of generality, to the case where the vacuum on the left side of the wall is the $0^{\text{th}}$ one. We then call $k$-wall, with $0 < k < h$, a wall that connects the $0^{\text{th}}$ vacuum to the $k^{\text{th}}$ vacuum. Formula (2.5) shows that a system of separated parallel BPS domain walls is unstable towards forming a unique domain wall in which the phase of the gaugino condensate jumps by the total amount, because the tension of a $k$-wall is lower than $k$ times the tension of a 1-wall. Equivalently, parallel BPS domain walls have central charges with different phases and thus are not mutually BPS.

Another useful property is that the 3d physics on an $(h-k)$-wall is the parity reversal of that on a $k$-wall. Indeed, we can perform a rotation by $\pi$ in a plane formed by the direction orthogonal to the wall and a direction along the wall. The resulting configuration connects the $k^{\text{th}}$ vacuum on the left to the $0^{\text{th}}$ vacuum on the right, which is equivalent to an $(h-k)$-wall, with one direction along the wall being inverted.

One is interested in studying the existence, degeneracy and other features of the BPS domain walls of SYM. This question has been analyzed in great detail by Acharya and Vafa [42].

---

[3]In particular, the generator of the $\mathbb{Z}_2$ subgroup of $\mathbb{Z}_{2h}$ corresponds to a spatial rotation by $2\pi$.

Specifically, in the case $G = SU(N)$, AV employed a brane construction to provide a 3d world-volume theory that describes the domain wall dynamics. One can realize 4d $\mathcal{N} = 1$ $SU(N)$ SYM using a $G_2$-holonomy geometry in M-theory [58]. Such a seven-dimensional manifold is a $\mathbb{Z}_N$ quotient of the spin bundle on $S^3$, topologically $(S^3 \times \mathbb{R}^4)/\mathbb{Z}_N$. The $\mathbb{Z}_N$ acts differently in the UV and in the IR, in a way which is continuous in the quantum theory and that provides an M-theory version of the geometric transition [59]. In the IR, it acts freely on $S^3$ producing the spin bundle on the lens space $S^3/\mathbb{Z}_N$. By reducing to type IIA along the Hopf fiber of the lens space, one obtains a resolved conifold geometry with $N$ units of RR $F_2$ flux through the blown-up $S^2$. Domain walls are realized by M5-branes wrapping the $S^3/\mathbb{Z}_N$ in M-theory. In particular, $k$ M5-branes shift the vacuum by $k$ units and realize a $k$-wall. In type IIA they reduce to $k$ D4-branes wrapping the two-sphere. Taking into account the Wess-Zumino coupling to the RR background, the domain wall worldvolume theory is the 3d $\mathcal{N} = 2$ $U(k)$ gauge theory, with an $\mathcal{N} = 1$ Chern-Simons interaction that reduces the supersymmetry. Using $\mathcal{N} = 1$ notation, the theory is[4]

$$\text{3d } \mathcal{N} = 1 \ U(k)_N \text{ gauge theory with a (singlet + adjoint) scalar multiplet .} \tag{2.6}$$

The singlet is decoupled and free at low energies. It is the Goldstone mode associated to broken translations (and fermionic partners) and it describes the center-of-mass motion of the domain wall perpendicular to its worldvolume. It can only have derivative couplings with the rest of the theory, and those are suppressed at low energy. The adjoint can describe the breaking of the $k$-wall into $k$ 1-walls. It has vanishing bare mass, producing a *classical* moduli space along which it has diagonal vacuum expectation values (VEVs): each entry represents the position, relative to the center of mass, of one of the 1-walls the $k$-wall breaks into. As previously noticed, though, it follows from (2.5) that quantum corrections lift the classical moduli space. If one is interested in the low-energy behavior, the adjoint scalar multiplet can be integrated out. A careful analysis [16, 60, 61] shows that the effective mass is negative.[5] We can thus use the alternative low-energy description

$$\text{3d } \mathcal{N} = 1 \ U(k)_{N-\frac{k}{2}, N} \text{ gauge theory .} \tag{2.7}$$

(Here and in the following we will neglect the decoupled and free center of mass.) This theory has a single supersymmetric gapped vacuum [62] in which the gaugino has negative mass. Integrating the gaugino out as well, at low energy we are left with a gapped vacuum hosting the topological (spin-)Chern-Simons theory

$$\text{3d } U(k)_{N-k, N} \ . \tag{2.8}$$

As it should be, one can check that the worldvolume theories on a $k$-wall and on an $(N-k)$-wall are related by parity reversal. This follows from the 3d IR duality

$$\mathcal{N} = 1 \quad U(k)_{N-\frac{k}{2}, N} \qquad \longleftrightarrow \qquad \mathcal{N} = 1 \quad U(N-k)_{-\frac{N+k}{2}, -N} \qquad \text{for } 0 \leq k \leq N, \tag{2.9}$$

which in turn reduces to the level-rank duality of CS theories $U(k)_{N-k, N} \longleftrightarrow U(N-k)_{-k, -N}$ [6, 7]. Notice that in the extremal case of an $N$-wall, the proposal (2.7) gives $\mathcal{N} = 1$ $U(N)_{\frac{N}{2}, N}$ which has a trivially gapped vacuum. This is consistent with the fact that an $N$-wall decays to the 4d vacuum.

---

[4]To avoid confusion, with "singlet" and "adjoint" we refer to the two irreducible representations. Together, they form a reducible representation that is usually called the adjoint representation of $U(k)$.

[5]What we mean is that the scalar components have positive squared mass, while the fermion components have negative mass.

One of the implications of the string theory construction is that on flat Minkowski space, in each soliton sector there is a single BPS $k$-wall. This corresponds to the fact that the worldvolume theory (2.7) has a single gapped vacuum on the spatial manifold $\mathbb{R}^2$. On the other hand, in the presence of topological sectors, the vacuum degeneracy can change as we change the spatial topology. The net number of vacuum states—weighted by the fermion number $(-1)^F$—on $T^2$ with periodic boundary conditions for fermions is captured by the Witten index. For the theory (2.7) on a $k$-wall the Witten index is

$$\mathrm{WI}\Big[U(k)_{N-\frac{k}{2},N}\Big] = \binom{N}{k}. \tag{2.10}$$

This corresponds to the number of fermionic lines of the spin-TQFT (2.8).[6] The Witten index matches the net number of domain walls one observes in the system dimensionally reduced on $T^2$ down to two dimensions [42].

## 2.1 Interface operators

According to (2.8), the $k = 1$ domain wall is described at low energy by a $U(1)_N$ Chern-Simons TQFT, which is level/rank dual to an $SU(N)_{-1}$ TQFT. Keeping $\mathcal{N} = 1$ supersymmetry manifest, the latter is an $\mathcal{N} = 1$ $SU(N)_{-1-\frac{N}{2}}$ CS theory. We would like to show that its action can be reproduced in the IR by a different procedure: by inserting an interface operator that interpolates between $\theta$ and $\theta + 2\pi$ as we move along one spatial direction, say $x_3$. We stress that the interface operator is not a dynamical excitation of the system, and it corresponds instead to an explicit deformation of the theory [2] (for instance, it does not lead to Goldstone modes).

Let us then consider the SYM action with a space-dependent $\theta$ angle, interpolating between a value $\theta$ at $x_3 \to -\infty$ and $\theta + 2\pi$ at $x_3 \to +\infty$. Eventually, we will take an IR limit in which $\theta(x_3)$ becomes a step function localized at $x_3 = 0$. The space dependence has two effects. First, the SYM action is not supersymmetric anymore. It is possible to preserve half of the supercharges by adding an extra term, therefore the interface operator is 1/2 BPS, like BPS domain walls. Second, as we take the IR limit, the interface operator induces a bare $\mathcal{N} = 1$ Chern-Simons term at level $-1$, including the correct gaugino mass term, along the 3d surface $x_3 = 0$. This is precisely the bare action of $\mathcal{N} = 1$ $SU(N)_{-1-\frac{N}{2}}$ CS theory (while the contribution $-\frac{N}{2}$ to the CS term comes from the 1-loop regularization of the gaugini).

We stress that the computation that follows is not limited to gauge group $SU(N)$: it can be repeated verbatim for any gauge group $G$, including exceptional and product groups.

Let us start considering the action of SYM. Neglecting the auxiliary fields $D^A$, which vanish on-shell, it reads

$$S_{\mathrm{SYM}} = \int d^4x \left[ -\frac{1}{4g^2} F_A^{\mu\nu} F_{\mu\nu}^A + \frac{\theta}{32\pi^2} \epsilon_{\mu\nu\rho\sigma} F_A^{\mu\nu} F^{\rho\sigma A} + \frac{i}{2g^2} \bar{\lambda}_A \gamma^\mu D_\mu \lambda^A \right]. \tag{2.11}$$

We use four-component spinor notation, and follow the conventions of appendix A of [63]. The supersymmetry variations are

$$\delta A_\mu^A = -i\,\bar{\epsilon}\gamma_\mu \lambda^A, \qquad \delta\lambda^A = \frac{1}{2} F_{\mu\nu}^A \gamma^{\mu\nu} \epsilon. \tag{2.12}$$

If the $\theta$ angle is constant on space-time, the action (2.11) is invariant. If, instead, we take it to be a function of the spatial coordinate $x_3$, we have

$$\delta S = \int d^4x \left[ -i\,\frac{\partial^3 \theta(x_3)}{8\pi^2} \epsilon_{3\nu\rho\sigma} \bar{\lambda}_A \gamma^\nu \epsilon F^{\rho\sigma A} \right] \neq 0. \tag{2.13}$$

---

[6]States on $T^2$ with periodic boundary conditions for fermions are prepared by the path-integral on a solid torus with a fermionic line at its core.

We can make the SYM action with varying $\theta$ angle invariant under half of the supersymmetries by adding the term

$$S_{\text{varying } \theta} = \int d^4x \left[ i \frac{\partial^3 \theta(x_3)}{32\pi^2} \bar{\lambda}_A \left(1 + \gamma^0 \gamma^1 \gamma^2\right) \lambda^A \right] \tag{2.14}$$

and imposing the constraint

$$\left(1 - \gamma^0 \gamma^1 \gamma^2\right) \epsilon = 0 \tag{2.15}$$

to the supersymmetry parameter $\epsilon$.

We are interested in configurations in which the $\theta$ angle varies by $2\pi$ from $x_3 = -\infty$ to $x_3 = +\infty$. Let us now consider the IR limit in which $\partial_3 \theta(x_3) = 2\pi \delta(x_3)$. Integrating by parts the $\theta$-term in (2.11) and neglecting boundary terms at infinity, we obtain

$$\frac{1}{8\pi^2} \int_{\mathcal{M}_4} \theta(x_3) \operatorname{Tr} F \wedge F = -\frac{1}{8\pi^2} \int_{\mathcal{M}_4} 2\pi \delta(x_3) \, dx_3 \wedge \operatorname{Tr}\left(A \wedge dA - \tfrac{2i}{3} A^3\right)$$
$$= -\frac{1}{4\pi} \int_{\mathcal{M}_3} \operatorname{Tr}\left(A \wedge dA - \tfrac{2i}{3} A^3\right). \tag{2.16}$$

Here $\mathcal{M}_4$ is the 4d space-time manifold, $\mathcal{M}_3$ is the 3d location of the interface operator at $x_3 = 0$, and we used standard conventions to rewrite the $\theta$-term using differential forms, e.g. $F = \frac{1}{2} F_{\mu\nu} dx^\mu \wedge dx^\nu = dA - iA \wedge A$ and $\operatorname{Tr}(T^A T^B) = \delta^{AB}$. We see that, with a suitable choice of the induced orientation, the interface operator has a 3d worldvolume action that includes a bare $SU(N)$ CS term at level $-1$.

In the same IR limit, also (2.14) can be recast as a genuine 3d term, specifically as a 3d gaugino mass term. The 4d gaugino $\lambda^A$ is a Majorana spinor and, using the conjugation matrix $\begin{pmatrix} i\sigma_2 & 0 \\ 0 & -i\sigma_2 \end{pmatrix}$, it can be written as

$$\lambda^A = \begin{pmatrix} \xi \\ i\sigma_2 \xi^* \end{pmatrix}, \tag{2.17}$$

where $\xi$ is a two-component spinor. Defining now the 3d spinor $\chi = \frac{1}{2}(\xi + \sigma_1 \xi^*)$, we can write (2.14) as

$$S_{\text{varying } \theta} = \frac{1}{4\pi} \int_{\mathcal{M}_3} d^3x \, \operatorname{Tr} \bar{\chi} \chi, \tag{2.18}$$

where $\bar{\chi} = \chi^\dagger i\sigma_2$ is the 3d conjugate spinor. Putting together the above equation with (2.16), we get a bare $\mathcal{N} = 1$ CS term at level $-1$.

## 3 Domain walls of $SU(N)$ SQCD

Let us now move to the theory of interest, namely 4d $\mathcal{N} = 1$ $SU(N)$ SQCD with $F$ flavors, described by $F$ chiral superfields $Q$ and $\widetilde{Q}$ in the fundamental and antifundamental representation, respectively. This theory exhibits very different low-energy physics depending on $N$ and $F$ [21–25] (see also the review [26]). In this paper we study the case $F < N$.[7]

If quarks are massless, the theory has runaway behaviour and no stable vacua [23]. We thus study the theory with massive quarks. We choose a diagonal superpotential mass term that preserves a diagonal $SU(F)$ subgroup of the original $SU(F)_L \times SU(F)_R$ chiral flavor symmetry. Besides, the theory has a baryonic $U(1)_B$ symmetry[8] (that will play no rôle) as well as

---

[7]SQCD with $F \geq N$ has a quantum exact moduli space which includes both mesonic and baryonic VEVs, and which requires a somewhat different treatment. Domain walls in SQCD with $F \geq N$ will be discussed elsewhere [27].

[8]In the special case of $SU(2)$ massive SQCD, the symmetry $SU(F) \times U(1)_B$ is enhanced to $Sp(F)$.

a $\mathbb{Z}_{2N}$ R-symmetry under which the flavor superfields have charge 1. The vacua of the theory are determined by considering the effective superpotential $W$ on the space of VEVs of the gauge-invariant meson superfield $M = \widetilde{Q}Q$, which is an $F \times F$ matrix transforming in the adjoint representation of the $SU(F)$ flavor symmetry. The effective superpotential gets contribution from the bare mass term and from the non-perturbatively generated Affleck-Dine-Seiberg (ADS) superpotential [23]:

$$W = m_{4d} \operatorname{Tr} M + (N - F)\left(\frac{\Lambda^{3N-F}}{\det M}\right)^{\frac{1}{N-F}} . \tag{3.1}$$

This gives $N$ gapped vacua, with

$$M = \widetilde{M} \, \mathbb{1}_F , \qquad \widetilde{M}^N = \frac{\Lambda^{3N-F}}{m_{4d}^{N-F}} , \tag{3.2}$$

corresponding to the spontaneous breaking $\mathbb{Z}_{2N} \to \mathbb{Z}_2$. The gaugino condensate can be obtained integrating in the glueball superfield, or directly differentiating $W$ with respect to $\log \Lambda^{3N-F}$: $\langle \lambda\lambda \rangle = m_{4d}\widetilde{M}$.

Gapped vacua can be separated by domain walls, possibly half-BPS. We are interested in determining the low-energy worldvolume theory on these domain walls, from which their properties can be inferred.

For large values of the quark mass $m_{4d}$ (compared to $\Lambda$), flavors can be integrated out leaving $SU(N)$ SYM at low energy. In this regime, the domain walls must be described by the worldvolume theory (2.7). When the mass $m_{4d}$ becomes much smaller than $\Lambda$, instead, one could expect the dynamics to be different. As we will discuss in Section 5, in this limit the SQCD vacua fly to large expectation values of $M$ where a Higgsed description is appropriate, and domain walls connecting the vacua can be reliably described semi-classically. In this regime their dynamics and vacuum structure look in fact much different from those in pure SYM. In particular, we will see that there exist multiple degenerate walls connecting the same vacua. We thus expect interesting phase transitions connecting the large and small mass regimes. This resembles what happens in massive QCD, as recently discussed in [2, 3] (and in [5] within a holographic context). The three-dimensional worldvolume theory we are after should reproduce all such features.

Note that, as long as the quark masses are non-vanishing, there are no flat directions in field space and the Witten index of the domain wall worldvolume theory cannot jump. Hence, the Witten index must be $\binom{N}{k}$ as in SYM. There exists an extensive literature on BPS domain walls in SQCD, which includes [28–48]. We notice that the existing lists cannot be complete because, in general, they do not reproduce the Witten index (2.10). Our proposal will fill this gap.

## 4 Three-dimensional worldvolume theory

We cannot rigorously derive the worldvolume theories on domain walls, but we can get some intuition about what those theories should look like by extending the Acharya-Vafa brane construction from SYM to SQCD. In the type IIA string theory setting, flavors can be added introducing $F$ D6-branes extending in the four space-time directions supporting the gauge theory, and wrapping a non-compact special Lagrangian three-cycle of the resolved conifold (after the geometric transition) [64]. Such a three-cycle is an $\mathbb{R}^2$ bundle over the equatorial $S^1$ inside the blown-up $S^2$. Together with the $N$ color D6-branes which, through the geometric transition, get replaced by $N$ units of RR $F_2$ flux on $S^2$, the branes realize at low energy 4d $\mathcal{N} = 1$ $SU(N)$

SQCD with $F$ flavors and a quartic superpotential. Flavor masses correspond to the D6-branes reaching a minimal radial position $r_0 \sim m_{4d}$. This is not quite the theory we are interested in, since SQCD with quartic superpotential has a different number of vacua from the theory without it, but we can still use it to get some intuition about the domain wall theories.

Domain walls correspond to D4-branes wrapped on the blown-up $S^2$ at the tip of the conifold, as in Section 2. However, the presence of the flavor D6-branes gives rise to a new open string sector at the intersection. This suggests that the 3d $\mathcal{N} = 1$ domain wall theory should contain $F$ scalar multiplets in the fundamental representation. Moreover, there should not be bare superpotential couplings involving the (singlet + adjoint) scalar multiplet $\Phi$, as in (2.6), because the singlet becomes at low energy the free and decoupled center-of-mass superfield, while the diagonal components of the adjoint describe the breaking of a $k$-wall into 1-walls and should be flat directions for large VEVs.

We will not push the similarity any further, since we are interested in SQCD without quartic superpotential, and instead propose that the effective theory on $k$-walls be[9]

$$3d \ \mathcal{N} = 1 \ U(k)_{N-\frac{F}{2}} \text{ gauge theory with a (singlet + adjoint) scalar multiplet } \Phi$$
$$\text{and } F \text{ fundamental scalar multiplets } X \ , \tag{4.1}$$

and no bare superpotential involving $\Phi$. We expect the bare 3d mass of $X$ to be proportional to the 4d mass $m_{4d}$. As in Section 2, the singlet is the Goldstone mode associated to broken translations (and fermionic partners) and will be neglected in the following. The adjoint classically gives rise to flat directions, which however are lifted by quantum effects. The one-loop computation of [16, 18, 60, 61] is still valid for $\Phi$ since the latter has no bare superpotential couplings, and it leads to negative mass around $\Phi = 0$, as expected from the four-dimensional brane charge. Integrating out the adjoint[10] we obtain the simpler low-energy description

$$3d \ \mathcal{N} = 1 \ U(k)_{N-\frac{k}{2}-\frac{F}{2}, N-\frac{F}{2}} \text{ with } F \text{ flavors } X \ . \tag{4.2}$$

Renormalization effects change the three-dimensional mass and produce quartic superpotential terms (which are classically marginal):

$$\mathcal{W} = \frac{1}{4} \text{Tr} X^\dagger X X^\dagger X + \frac{\alpha}{4} \left( \text{Tr} X^\dagger X \right)^2 + \frac{m}{2} \text{Tr} X^\dagger X \ . \tag{4.3}$$

Notice that there are two independent quartic gauge-invariant combinations. The overall scale has been arbitrarily fixed for convenience. The relative signs (with respect to the sign of the Chern-Simons term) instead are physical and have been fixed in order to reproduce the expected behavior at large and small (compared with $\Lambda$) 4d mass $m_{4d}$. Consistency also requires that $\alpha > -\min(k, F)^{-1}$. The 3d parameter $m$ is an effective IR mass: as we will see, at $m = 0$ there is a second-order phase transition. Although we do not know the precise relation between $m_{4d}$ and $m$, we will see that large values of $m_{4d}$ correspond to $m > 0$ and small values of $m_{4d}$ correspond to $m < 0$. We will indicate as $m_{4d}^*$ the value that corresponds to $m = 0$. Higher order terms in $\mathcal{W}$ are expected to be irrelevant at the point $m = 0$.

In the remainder of this section, we will study the dynamics of the theory (4.2)-(4.3) on its own. Later, in Section 5, we will confront it with actual massive SQCD domain wall dynamics.

Let us note, from the outset, that our proposal already satisfies an important consistency check. The theory (4.2) enjoys the following $\mathcal{N} = 1$ infrared duality:

$$U(k)_{N-\frac{k+F}{2}, N-\frac{F}{2}} \text{ with } F \text{ flavors } \quad \longleftrightarrow \quad U(N-k)_{-\frac{N+k-F}{2}, -N+\frac{F}{2}} \text{ with } F \text{ flavors } , \tag{4.4}$$

---

[9]See footnote 4.

[10]The discussion that follows is valid for vacua in which $\Phi = 0$. Comparing with [16], it seems unlikely that the theory with vanishing bare mass for $\Phi$ has other vacua in which $\Phi$ gets an expectation value, and the Witten index supports this claim. However, we cannot rigorously exclude it. We leave a more detailed analysis of the theory with both the adjoint and the fundamental fields for future work.

where on both sides there are quartic $\mathcal{N} = 1$ superpotentials. This duality was discovered in [18] in a very similar context.[11] The authors consider the theory (4.2) with quadratic but not quartic UV bare superpotential, argue that there exists a value of the bare mass for which the theory flows to an $\mathcal{N} = 1$ fixed point, and conjecture that the two theories in (4.4) lead to the very same fixed point. Our claim is that the effective theory at the fixed point has superpotential (4.3) with $m = 0$. Such an effective description will allow us to use a semi-classical analysis to understand the relevant deformation triggered by the mass term. For instance, following [18] we will show that massive vacua match. The duality (4.4) is a strong consistency check of our proposal, since it relates $k$-walls to time-reversed $(N - k)$-walls. As already emphasized, this is an expected feature of $k$-walls in SQCD.

Let us notice another interesting fact. For $k = 1$, the proposed domain wall theory enjoys another $\mathcal{N} = 1$ duality [18]:

$$U(1)_{N-\frac{F}{2}} \text{ with } F \text{ flavors} \quad \longleftrightarrow \quad SU(N)_{-1-\frac{N-F}{2}} \text{ with } F \text{ flavors} , \qquad (4.5)$$

with quartic superpotential on both sides (we propose in Section 4.2 that the theory on the left has emergent $\mathcal{N} = 2$ supersymmetry in the IR, and hence the same should happen to the theory on the right). Intriguingly, the gauge group is the same as that of the four-dimensional theory. This suggests that the theory on the right might be reproduced by an interface operator as discussed in Section 2.1 for pure SYM (and in [2,3] without supersymmetry): the contribution $-1$ to the CS level could come from an $x$-dependent theta angle, $-\frac{N}{2}$ from the regularized 1-loop determinant of gaugini, and $\frac{F}{2}$ from the flavors. It would be interesting to make this idea concrete.

## 4.1 Analysis of vacua

Let us study the vacua of the three-dimensional theory (4.2) with superpotential (4.3), where we assume $\alpha > -\min(k, F)^{-1}$. The scalar superfields are $k \times F$ matrices $X_{ai}$, with $a$ and $i$ being gauge and flavor indices, respectively. The F-term equation is

$$0 = XX^\dagger X + \alpha \left( \text{Tr} X^\dagger X \right) X + mX . \qquad (4.6)$$

By gauge and flavor rotations, we can bring $X$ to a rectangular diagonal form. In this basis, both $XX^\dagger$ and $X^\dagger X$ are diagonal with real non-negative entries: they have $\min(k, F)$ eigenvalues in common, that we indicate by $\lambda_j \geq 0$, while the remaining eigenvalues of the larger of the two matrices vanish.

The eigenvalues have to satisfy the equations

$$\lambda_j^2 + \alpha \left( \sum_i \lambda_i \right) \lambda_j = -m\lambda_j \qquad j = 1, \ldots, \min(k, F) . \qquad (4.7)$$

For $m \neq 0$, up to permutations these equations have $\min(k, F) + 1$ solutions, that we parametrize by $J = 0, \ldots, \min(k, F)$. Each solution has only $J$ non-vanishing (and identical) eigenvalues:

$$\text{solution } J : \qquad \lambda_1, \ldots, \lambda_J = \frac{-m}{1 + J\alpha} , \qquad \lambda_{J+1}, \ldots, \lambda_{\min(k,F)} = 0 . \qquad (4.8)$$

The solutions with $J > 0$ are acceptable only if $-m/(1 + J\alpha)$ is non-negative. This gives a different number of vacua for $m$ positive and negative.

---

[11]The case $F = 1$ was discussed in [16]. However, as we explain in Section 4.2, the CFT at the phase transition is conjectured to have emergent $\mathcal{N} = 2$ supersymmetry and, if this is the case, the duality is the one already found in [65].

• **$m > 0$.** Only the vacuum with $J = 0$, in which $X = 0$, is acceptable. In such a vacuum, quarks have positive mass and can be integrated out, leaving

$$\text{3d} \quad \mathcal{N} = 1 \quad U(k)_{N-\frac{k}{2},N} . \tag{4.9}$$

Since $k < N$, this theory has a single supersymmetric gapped vacuum that hosts the TQFT $U(k)_{N-k,N}$, and its Witten index is

$$\text{WI} = \binom{N}{k} . \tag{4.10}$$

This is the expected result for the behavior of SQCD domain walls in the large 4d mass regime, $m_{4\text{d}} \gg \Lambda$, as discussed in Section 3.

• **$m < 0$.** All $\big(\min(k,F)+1\big)$ vacua, labelled by $J$, are acceptable. The quark field $X$ gets a VEV, which can be brought to a diagonal rectangular form with $J$ non-vanishing identical entries (for $J = 0$ the VEV is zero). This breaks the flavor symmetry as

$$SU(F) \to S\big[U(J) \times U(F-J)\big] , \tag{4.11}$$

leading to a supersymmetric NLSM in the IR with target space $U(F)/\big(U(J) \times U(F-J)\big)$ (for $J = 0$ and $J = F$ the symmetry is not broken). All other fields become massive, either because of the potential or because of Higgs mechanism. Indeed, the VEV also breaks the gauge symmetry as $U(k) \to U(k-J)$. Fermions charged under the unbroken gauge group, coming from the quark superfields and from the broken components of the gaugino, mix and become massive as well. In particular, $F$ eigenmodes get a negative mass and $J$ get a positive mass.[12] All such modes transform in the fundamental representation of $U(k-J)$. As a result, the bare CS level of the unbroken gauge group is shifted by $-F$. This leads to an $\mathcal{N} = 1$ pure gauge CS theory $U(k-J)_{N-F-\frac{k-J}{2},N-F}$ in the IR (for $k = J$ this factor is not present). The two factors are decoupled in the IR, thus the low energy theory around a vacuum labelled by $J$ is

$$\mathcal{N} = 1 \qquad U(k-J)_{N-F-\frac{k-J}{2},N-F} \quad \times \quad \text{NLSM} \quad \frac{U(F)}{U(J) \times U(F-J)} . \tag{4.12}$$

The supersymmetric NLSM has a Wess-Zumino term, which is conveniently specified by describing the NLSM as an $\mathcal{N} = 1$ $U(J)_{N-\frac{J+F}{2},N-\frac{F}{2}}$ gauge theory coupled to $F$ fundamental scalar multiplets getting VEV. Notice in passing that the NLSM target is a Kähler manifold—the complex Grassmannian—and thus if we truncate the effective Lagrangian at two-derivative level, the NLSM has emergent 3d $\mathcal{N} = 2$ supersymmetry [45].

The gauge theory on the left of (4.12) has Witten index $\text{WI} = \binom{N-F}{k-J}$, which vanishes for $N - F - k + J < 0$ (recall that $N > F$ and $k \geq J$). Indeed, the theory breaks supersymmetry in that regime. This is a non-perturbative effect that lifts some of the would-be vacua labelled by $J$. Eventually, supersymmetric vacua correspond to

$$\max(0, F+k-N) \leq J \leq \min(k,F) . \tag{4.13}$$

In each supersymetric vacuum, the Witten index of the low-energy theory is

$$\text{WI} = \binom{N-F}{k-J}\binom{F}{J} , \tag{4.14}$$

---

[12] The components of the $F$ flavors charged under the unbroken gauge group are not coupled to the scalars getting VEV, thus they have a mass term $-m$ from the superpotential. However there are also mixed mass terms with the $J$ components of the gaugino along "block-off-diagonal" broken generators, from the Yukawa couplings imposed by supersymmetry. Finally, there is a gaugino mass term from the supersymmetrization of the CS term. The analysis is similar to the one in [17].

which is positive. Using the binomial identity[13]

$$\sum_{J=\max(0,F+k-N)}^{\min(k,F)} \binom{N-F}{k-J}\binom{F}{J} = \binom{N}{k} \qquad \text{for } N \geq F , \qquad (4.15)$$

we see that the total Witten index at $m < 0$ agrees with the one at $m > 0$.

At $m = 0$ there is a phase transition in which the multiple vacua at $m < 0$ simultaneously coalesce into the single vacuum at $m > 0$. Such a phase transition—essentially because of supersymmetry—is necessarily second order and thus it is described by a 3d $\mathcal{N} = 1$ SCFT. To understand this point, already stressed in [16, 18], notice the following facts. First, in our range of parameters, the number $\big(\min(k,F) - \max(0,F+k-N) + 1\big)$ of vacua at $m < 0$ is always greater than one, while at $m > 0$ there is a single vacuum. Second, each of those vacua has positive Witten index (4.14) or (4.10). Vacua with non-vanishing Witten index must necessarily have zero energy: they cannot change their vacuum energy in isolation, the only way is to pair with other vacuum states so that the total Witten index is zero. Therefore the multiple vacua at $m < 0$ must coalesce at the phase transition, which cannot be first order and must be second (or higher) order. Third, solving the F-term equations derived from (4.3), we found that all vacua at $m < 0$ coalesce simultaneously. This conclusion is not modified if we arbitrarily perturb (4.3) with higher-order terms, and thus remains true to all orders in perturbation theory. The $\mathcal{N} = 1$ SCFT at $m = 0$ is the one that enjoys the IR duality (4.4). Let us stress once more that while we have not determined the precise relation between 3d and 4d masses, the value $m = 0$ corresponds to some value $m_{4d}^*$ of the 4d mass, of order the dynamically generated SQCD scale $\Lambda$. We expect $m_{4d}^*$ to depend on $N, F, k$.

It is useful to organize the different vacua—as we vary $J$—of the various 3d domain wall theories—as we vary $k$—for fixed 4d theory (namely for fixed $N, F$) into a table. Let us indicate the $\mathcal{N} = 1$ NLSM with target the complex Grassmannian by

$$\text{Gr}(J,F) = \frac{U(F)}{U(J) \times U(F-J)} = \text{Gr}(F-J,F) . \qquad (4.16)$$

In Table 1 we put the NLSMs $\text{Gr}(J,F)$ with $0 \leq J \leq F$ on the horizontal axis, and the topological sectors $\mathcal{N} = 1 \ U(\Delta)_{N-F-\frac{\Delta}{2},N-F}$ with $0 \leq \Delta \leq N-F$ on the vertical axis. The list of vacua for one theory with given $k$ are read diagonally (along a line from bottom-left to upper-right) and the corresponding values of $J$ are read in the last row. In the table we have also specified the level-rank duality of spin-CS theories [7], expressed in $\mathcal{N} = 1$ notation by (2.9). We have already observed that, employing the IR duality (4.4), the worldvolume theory on $k$-walls is the parity reversal of the theory on $(N-k)$-walls. Here we can consistently check, as already done in [18], that also their vacua have the same property. In particular, a vacuum of $k$-wall labelled by $J$ is the parity reversal of a vacuum of $(N-k)$-wall labelled by $F-J$.

As manifest in Table 1, some vacua are special. The ones in the first and last column do not break the flavor symmetry and thus are fully gapped without massless Goldstone fields. We call them "symmetry preserving walls". The ones in the first and last row, instead, do not host a topological sector.

In Section 5 we will construct domain walls as BPS codimension-one solitons interpolating between the $N$ vacua of four-dimensional massive SQCD in the regime of small $m_{4d}$, finding perfect agreement with the 3d dynamics discussed above.

---

[13]To derive the identity, start with $(1+x)^n(1+x)^m = (1+x)^{n+m}$ with $n,m \geq 0$ and apply the binomial expansion $(1+x)^n = \sum_{j=0}^n \binom{n}{j}x^j$. Equating the coefficients gives $\sum_{j=\max(0,s-n)}^{\min(m,s)} \binom{n}{s-j}\binom{m}{j} = \binom{n+m}{s}$.

Table 1: Domain walls of massive $SU(N)$ SQCD with $F < N$ flavors and small $m_{4d}$, for all values of $k$. We gather the vacua of the conjectured 3d worldvolume theories in the regime $m < 0$, as $k$ and $J$ are varied, into a table of size $(N - F + 1) \times (F + 1)$. For each vacuum, the low energy theory is the product of a topological sector (on the vertical axis in $\mathcal{N} = 1$ notation) and an $\mathcal{N} = 1$ NLSM (on the horizontal axis). The different vacua at fixed $k$ are read diagonally, and the corresponding value of the label $J$ is in the last row. The two empty cells correspond formally to $k = 0$ and $k = N$.

| | Gr$(0,F)$ trivial | Gr$(1,F) \leftrightarrow$ Gr$(F-1,F)$ | Gr$(2,F) \leftrightarrow$ Gr$(F-2,F)$ | $\cdots$ | Gr$(F-1,F)$ $\leftrightarrow$ Gr$(1,F)$ | Gr$(F,F)$ trivial |
|---|---|---|---|---|---|---|
| trivial $U(N-F)_{\frac{F-N}{2},F-N}$ | | $k=1$ | $k=2$ | $\cdots$ | $k=F-1$ | $k=F$ |
| $U(1)_{N-F} \leftrightarrow$ $U(N-F-1)_{\frac{F-N-1}{2},F-N}$ | $k=1$ | $k=2$ | $k=3$ | $\cdots$ | $k=F$ | $k=F+1$ |
| $U(2)_{N-F-1,N-F} \leftrightarrow$ $U(N-F-2)_{\frac{F-N-2}{2},F-N}$ | $k=2$ | $k=3$ | $k=4$ | $\cdots$ | $k=F+1$ | $k=F+2$ |
| $\vdots$ | $\vdots$ | $\vdots$ | $\vdots$ | $\ddots$ | $\vdots$ | $\vdots$ |
| $U(N-F-1)_{\frac{N-F+1}{2},N-F}$ $\leftrightarrow U(1)_{F-N}$ | $k=N-F-1$ | $k=N-F$ | $k=N-F+1$ | $\cdots$ | $k=N-2$ | $k=N-1$ |
| $U(N-F)_{\frac{N-F}{2},N-F}$ trivial | $k=N-F$ | $k=N-F+1$ | $k=N-F+2$ | $\cdots$ | $k=N-1$ | |
| | $J=0$ | $J=1$ | $J=2$ | $\cdots$ | $J=F-1$ | $J=F$ |

## 4.2 Supersymmetry enhancement

Let us end this section with the following interesting observation. For special values of $N$, $F$ and $k$, the domain wall theories at the value $m_{4d}^*$ of the 4d flavor mass that corresponds to the 3d second-order phase transition, can exhibit IR enhancement of the 3d superconformal symmetry from $\mathcal{N} = 1$ to $\mathcal{N} = 2$ or $\mathcal{N} = 4$.[14] More precisely, only for the "reduced" worldvolume theory (4.2)-(4.3) we conjecture supersymmetry enhancement, while the Goldstone boson for broken translations and a massless Majorana fermion still combine into an $\mathcal{N} = 1$ real scalar multiplet.

When $k = 1$ or $F = 1$, the two quartic superpotential terms in (4.3) are equal: there exists only one quartic term one can write compatible with all symmetries. The coefficient of that term is not fixed by $\mathcal{N} = 1$ supersymmetry, and since that coupling is classically marginal, it runs under RG flow. If the coupling coefficient is appropriately tuned, though, the massless theory has $\mathcal{N} = 2$ supersymmetry.[15] This can be seen by starting with the $\mathcal{N} = 2$ YM-CS gauge theory with $F$ chiral multiplets in the fundamental representation. There is no $\mathcal{N} = 2$ (holomorphic) superpotential we can write. Using $\mathcal{N} = 1$ notation, though, there is a superpotential

---

[14]In [45], as we mention after (4.12), it was observed that the effective theories in the vacua at $m < 0$, i.e. for $m_{4d} < m_{4d}^*$ below the phase transition point, have enhanced $\mathcal{N} = 2$ supersymmetry at low energy for all values of $N, F, k$. This is because the NLSM has Kähler target space, and the CS gauge theory is gapped. Our conjecture is much stronger: we claim supersymmetry enhancement at the interacting CFT point. Such a conjecture only applies to $k = 1$, $k = N - 1$ and $F = 1$.

[15]In fact, adding a mass term to the $\mathcal{N} = 1$ superpotential corresponds to turning on the real mass associated to the topological symmetry in $\mathcal{N} = 2$ notation. Therefore, also the massive theory has $\mathcal{N} = 2$ supersymmetry, at least at energies below $g_{\text{YM}}^2$ such that we are entitled to integrate the adjoint out.

$$\mathcal{W}_{\mathcal{N}=1} = g_{\text{YM}} \sum_{i=1}^{F} X_i^\dagger \Psi X_i - \frac{g_{\text{YM}}^2 h}{2} \operatorname{Tr} \Psi^2 \,, \tag{4.17}$$

where $\Psi$ is the adjoint $\mathcal{N}=1$ scalar superfield in the $\mathcal{N}=2$ vector multiplet, $g_{\text{YM}}$ is the Yang-Mills coupling and $h$ is the CS level. At energies below $g_{\text{YM}}^2$, the adjoint is non-dynamical and can be integrated out. This generates the quartic superpotential

$$\mathcal{W}_{\mathcal{N}=1} = \frac{1}{2h} \sum_{i,j=1}^{F} X_i^\dagger X_j \cdot X_j^\dagger X_i \,. \tag{4.18}$$

For $k=1$ or $F=1$, this is the same as $\mathcal{W}_{\mathcal{N}=1} = \frac{1}{2h}\big(\sum_i X_i^\dagger X_i\big)^2$, which is the only quartic term we can write. Thus, when the quartic term has coupling $1/2h$, the massless theory has $\mathcal{N}=2$ supersymmetry. For other values of the coupling, the supersymmetry is only $\mathcal{N}=1$, however one might suspect that the RG flow still drives the coupling to the $\mathcal{N}=2$ point in the IR, at least within a basin of attraction. Indeed, it has been shown in [18, 66, 67] that, at large CS level $h$, the $\mathcal{N}=2$ point is attractive.[16] It is very plausible, and we conjecture, that this is true even at small values of the CS level.

For values of the CS level such that the claim is true, the duality of fixed points (4.4) is really the $\mathcal{N}=2$ duality of "minimally chiral" theories found in [65]:

$$\mathcal{N}=2 \quad U(k)_{N-\frac{F}{2}} \text{ with } F \text{ fund.} \quad \longleftrightarrow \quad \mathcal{N}=2 \quad U(N-k)_{-N+\frac{F}{2}} \text{ with } F \text{ anti-fund,} \tag{4.19}$$

(valid for $F<N$) where here we take $k=1$ and/or $F=1$. The $\mathcal{N}=2$ superpotential vanishes on both sides. The duality implies that also for $k=N-1$ the $\mathcal{N}=2$ fixed point is attractive.

As noticed also in [18], there is no reason to expect supersymmetry enhancement at the fixed point in the other cases. The two quartic $\mathcal{N}=1$ superpotential terms are independent, giving rise to a two-dimensional RG flow. From (4.18), supersymmetry is enhanced to $\mathcal{N}=2$ when the coefficient of the single-trace term is $1/2h$ and that of the double-trace term is zero. One might suspect that, in this higher-dimensional RG space, stability of the $\mathcal{N}=2$ point gets lost. Indeed, it has been shown in [67] that, at large CS level, the $\mathcal{N}=2$ point has a repulsive direction in the two-dimensional space of quartic superpotential couplings, that ends up to an $\mathcal{N}=1$ point which is instead attractive.

The $\mathcal{N}=2$ duality is still useful, though: it turns out that all $\mathcal{N}=1$ dualities (4.4) (in their range $1<F<N$ and $1<k<N-1$) follow from an $\mathcal{N}=1$ quartic superpotential deformation of the $\mathcal{N}=2$ dualities (4.19). On the other hand, once we move outside the critical point of the phase transition, the low-energy theory is the product of a topological sector and a NLSM with Kähler target space: at two-derivative truncation this is an $\mathcal{N}=2$ theory for all values of $N$, $F$ and $k$.

When $k=1$ (or $k=N-1$, up to a parity transformation) we can obtain yet a different description using the $\mathcal{N}=2$ dualities of [68], namely

$$\mathcal{N}=2 \quad U(1)_{N-\frac{F}{2}} \text{ with } F \text{ fund.} \quad \longleftrightarrow \quad \mathcal{N}=2 \quad SU(N)_{-1-N+\frac{F}{2}} \text{ with } F \text{ anti-fund,} \tag{4.20}$$

with no $\mathcal{N}=2$ superpotential. Together, (4.19) at $k=1$ and (4.20) form a triality. As long as our conjecture about supersymmetry enhancement is correct, this triality is in fact the same as (4.4)-(4.5). Since we are considering $F<N$, the $\mathcal{N}=2$ SCFTs in (4.20) have trivial chiral ring and trivial moduli space of vacua (no BPS monopoles on the left, no BPS baryons on the right, and no mesons on either side).

---

[16]They also show that the basin of attraction includes the point with no quartic superpotential.

The case of $SU(2)$ SQCD with one flavor is doubly special because $k = N - k = 1$. In this case there are four dual domain wall theories:

$$\mathcal{N} = 1 \quad U(1)_{\pm\frac{3}{2}} \text{ with 1 flavor} \qquad \longleftrightarrow \qquad \mathcal{N} = 1 \quad SU(2)_{\pm\frac{3}{2}} \text{ with 1 flavor}. \qquad (4.21)$$

The critical point of the phase transition exhibit both enhanced supersymmetry and emergent time-reversal invariance. In $\mathcal{N} = 2$ language the above dualities become

$$\mathcal{N} = 2 \quad U(1)_{\pm\frac{3}{2}} \text{ with 1 flavor} \qquad \longleftrightarrow \qquad \mathcal{N} = 2 \quad SU(2)_{\pm\frac{5}{2}} \text{ with 1 flavor}, \qquad (4.22)$$

and were recently studied in [18, 69, 70]. In [71] it was argued that $\mathcal{N} = 2$ $U(1)_{\pm\frac{3}{2}}$ with one chiral flavor has infrared supersymmetry enhancement to $\mathcal{N} = 4$. The conclusion is that the BPS domain wall of 4d $\mathcal{N} = 1$ $SU(2)$ SQCD with 1 flavor is described by a 3d $\mathcal{N} = 4$ SCFT.

# 5 Four-dimensional constructions

The three-dimensional worldvolume theory (4.2) passes two non-trivial checks as a candidate for the effective theory describing massive SQCD domain walls. For large values $m_{4d} \gg \Lambda$ of the 4d mass, corresponding to positive 3d mass $m$ in our conventions, the theory reduces to (2.7) or equivalently (2.8), which describes domain walls in pure SYM. This is what one expects since, for large quark mass, SQCD reduces to pure SYM at low energy and so should the corresponding domain walls. A related check regards the Witten index, which remains constant along the phase transition at $m = 0$, see eqn. (4.15) and the discussion thereafter.

Our task in this section is to understand the regime of small 4d mass, $m_{4d} \ll \Lambda$. We will explicitly construct 1/2 BPS domain wall solutions in such a regime, and show that they precisely match the structure of multiple vacua of the three-dimensional worldvolume theory (4.2) with negative mass.

One of the key points which make our analysis possible is that for $m_{4d} \ll \Lambda$ the $N$ supersymmetric vacua of massive SQCD, eqns. (3.2), lie at large distance in the mesonic space, which is a Higgsed weakly-coupled region. Hence, the domain walls that interpolate between those vacua can be reliably constructed with a semi-classical analysis (up to an important caveat that we will discuss in the following).

In a weakly-coupled Wess-Zumino (WZ) model, domain walls can be constructed as finite-tension codimension-one solitonic configurations in which fields depend on one spatial coordinate, say $x$, and interpolate between the values in the two vacua at $x = \pm\infty$. For a standard WZ theory of chiral superfields $\Phi^a$ with two-derivative Lagrangian, described by a Kähler potential $\mathcal{K}(\Phi, \overline{\Phi})$ and a single-valued superpotential $W(\Phi)$, the domain wall equations are [56, 72]

$$\mathcal{K}_{a\bar{b}} \, \partial_x \Phi^a = e^{i\gamma} \, \partial_{\bar{b}} \overline{W}, \qquad (5.1)$$

where $\mathcal{K}_{a\bar{b}} = \partial_a \partial_{\bar{b}} \mathcal{K}$ are the non-vanishing components of the Kähler metric and $\gamma$ is the (constant) phase of the central charge of the domain wall, eqn. (2.3). It follows that

$$\partial_x W = e^{i\gamma} \mathcal{K}^{a\bar{b}} \frac{\partial W}{\partial \Phi^a} \frac{\partial \overline{W}}{\partial \overline{\Phi}^{\bar{b}}} = e^{i\gamma} \left\| \frac{\partial W}{\partial \Phi^a} \right\|^2, \qquad (5.2)$$

where, in the last expression, we have introduced a natural norm. Since the right-hand-side has constant phase, the image of $W(\Phi(x))$ is a straight line in the complex $W$-plane (and $e^{i\gamma}$ is its direction). The construction generalizes to cases where the superpotential $W(\Phi)$ is not a single-valued holomorphic function, but its derivatives are. The central charge is again the total excursion of the superpotential along $x$, eqn. (2.3).

When the WZ model includes a single chiral superfield, one can easily determine the existence of BPS domain walls. Let $W_{\pm\infty} = W\big(\Phi(x = \pm\infty)\big)$ be the values of the superpotential in the two vacua. One can invert $W(\Phi)$ and construct the pre-image of a straight line from $W_{-\infty}$ to $W_{+\infty}$. Such a pre-image will be made of one or more curves in the $\Phi$-plane (since $W$ is not an injective function, in general). Each curve that connects $\Phi(x = -\infty)$ to $\Phi(x = +\infty)$ identifies a BPS domain wall. On the other hand, we might not find any such curve. Note that this procedure only determines the orbit of $\Phi(x)$ in the complex $\Phi$-plane, not the precise profile of the field as a function of $x$. The latter depends on the Kähler potential $\mathcal{K}$.[17] However, as long as we are only interested in counting domain walls and determining their symmetry-breaking properties, this procedure suffices.[18] By contrast, in models with multiple chiral superfields $\Phi^a$ one should really solve the ODEs (5.1) in order to determine what types of domain walls exist and what their orbits are in field space. This can be done numerically, using shooting techniques.

In our case, the chiral superfields of the effective WZ model will be nothing but the components of the meson field. The meson matrix $M$ is proportional to the identity in supersymmetric vacua, see eqn. (3.2). If its evolution through the wall remains so, namely if its eigenvalues remain equal to one another, then we can reduce to one domain wall equation for a single chiral superfield $\widetilde{M}$, and the image of $\widetilde{M}(x)$ can be determined algebraically. If, instead, the eigenvalues split, and the meson matrix is not proportional to the identity along the wall, we have to resort to numerical analysis. In this case the domain wall breaks the $SU(F)$ flavor symmetry and thus its worldvolume theory includes Goldstone fields.

Before discussing these two classes of domain walls in more detail, let us address the caveat we have alluded to before.

**Domain walls in SYM.** The $N$ vacua of $SU(N)$ SYM and their gaugino condensate can be conveniently described using the Veneziano-Yankielowicz superpotential [73]

$$W_{\text{SYM}}(S) = S\left(\log\frac{\Lambda^{3N}}{S^N} + N\right). \tag{5.3}$$

Here $S \propto \text{Tr}\,\mathcal{W}^\alpha\mathcal{W}_\alpha$ is the gaugino superfield. The critical points and the value of the superpotential therein are

$$S = e^{\frac{2\pi i}{N}k}\Lambda^3, \qquad W\big|_S = e^{\frac{2\pi i}{N}k}N\Lambda^3, \tag{5.4}$$

with $k = 0, \ldots, N-1$.

One might then be tempted to use $W_{\text{SYM}}(S)$ as a standard WZ superpotential to construct domain walls interpolating between the $N$ vacua. However, this cannot be done for several related reasons. First, $W_{\text{SYM}}(S)$ is not the superpotential of a Wilsonian effective action for SYM, because $S$ does not describe the lightest particle. As a result, the superpotential is not a single-valued function of $S$. It is ambiguous by $2\pi i S\mathbb{Z}$, meaning that even its derivative is ambiguous by $2\pi i\mathbb{Z}$. Second, if $S$ winds once around the origin, $W_{\text{SYM}}$ shifts by $2\pi i N S$ which is not the minimal ambiguity. This means that the ambiguity is not resolved by going to a connected cover. The full domain of $W_{\text{SYM}}$ is made of $N$ disconnected components, each hosting one of the vacua. Thus, it is not possible to draw a continuous path from one vacuum to another. Third, $S$ is a constrained superfield because the imaginary part of its top component is the instanton density, whose integral is quantized. Thus, one should be careful in eliminating the auxiliary fields [74] and deriving the vacuum and domain wall equations. As a result, paths can effectively "jump" from one sheet to another, and this is not described within the semiclassical WZ theory. Papers dealing with these problems include [29, 43]. Very similar

---

[17]One should also assume that $\Phi(x)$ does not go through singularities of the Kähler metric.

[18]The general problem of counting domain walls was solved in [57].

problems arise when studying solitons in the 2d $\mathcal{N} = (2,2)$ $\mathbb{CP}^{N-1}$ model, using the effective theory on the Coulomb branch [75].

We will treat the domain walls of SYM as strongly-coupled BPS objects, with thickness of order $1/\Lambda$ and central charge given by the exact formula (2.3). As reviewed in Section 2, a $k$-wall across which the vacuum jumps as $S \to e^{2\pi i k/N} S$ hosts a topological sector described by an $\mathcal{N} = 1$ $U(k)_{N-\frac{k}{2}, N}$ theory, or, equivalently, a $U(k)_{N-k, N}$ CS theory.

**Domain walls in SQCD.** Contrary to the case of SYM, in $SU(N)$ SQCD with $F$ flavors there exists a weakly-coupled limit in which domain walls can be reliably constructed, *i.e.* the small mass regime $m_{4d} \ll \Lambda$. In this regime we can write a low-energy effective action for the mesons with superpotential (3.1) and Kähler potential

$$\mathcal{K} = 2 \operatorname{Tr} \sqrt{\overline{M} M} \,, \tag{5.5}$$

induced from the canonical one for quark superfields $Q, \widetilde{Q}$. The superpotential is in general multi-valued, but we can make it single-valued by working on a (connected) covering space of order $N - F$. We can then use such a WZ-like description to construct the domain walls. As long as the trajectories remain far away from the origin in field space, the WZ description is reliable.

For $F = N - 1$ this is the whole story. The superpotential is a single-valued function of $M$, the WZ model on the mesonic space is the Wilsonian low-energy effective action and all BPS domain walls are visible within such a description. The domain wall theory is either trivially gapped (besides the free decoupled center of mass), or it contains the Goldstone fields of a broken symmetry.

For $F < N - 1$, instead, at generic points on the mesonic space there is a residual SYM theory with gauge group $SU(N - F)$. Indeed, we can understand the non-perturbative ADS superpotential [23] as coming from gaugino condensation in the unbroken group. By scale matching we get

$$\Lambda^{3N-F} = \Lambda_{\text{unbroken}}^{3(N-F)} \det M \,. \tag{5.6}$$

Gaugino condensation gives $W_{\text{unbroken}} = (N-F)\big(\Lambda_{\text{unbroken}}^{3(N-F)}\big)^{1/(N-F)} = W_{\text{ADS}}$. The fact that the unbroken $SU(N-F)$ SYM has $N - F$ vacua leads to the multi-valuedness of the low energy superpotential. We can use this observation to construct two interesting classes of domain walls.

The simplest class of domain walls consists of configurations in which the vacuum of the unbroken $SU(N-F)$ is adiabatically evolved. This means that the domain wall profile connects the two vacua with a path in field space along which $W(M)$ is continuous. Because of (5.2), the path must be in the pre-image of a straight line with respect to the map $W(M)$. Such domain walls are essentially WZ walls, in which the $SU(N-F)$ gauge theory is a spectator. It follows that the worldvolume theory—besides the free and decoupled center of mass—is either trivially gapped, or it contains, again, the Goldstone fields of a broken symmetry.

A more subtle class of walls, that we call "hybrid", is obtained by combining a continuous evolution on the mesonic space with a shift of vacuum in the unbroken $SU(N-F)$. Such a shift implies that we transit from one sheet of the function $W(M)$ to another—according to the phase shift of the gaugino condensate in the unbroken gauge theory. Let us estimate the widths of the SQCD wall and of the transition in the unbroken SYM. The thickness of the SQCD wall is

$$\ell_{\text{SQCD}} \sim \frac{M}{\partial_x M} \sim \frac{M \, \mathcal{K}_{M\overline{M}}}{\partial W / \partial M} \sim \frac{1}{m_{4d}} \,, \tag{5.7}$$

where in the last equality we used the Kähler potential (5.5). Interestingly, in the $m_{4d} \to 0$ limit the domain wall size does not depend on the gauge dynamics. The thickness of the SYM

wall instead scales as $1/\Lambda_{\text{unbroken}}$. Using scale matching and the size of $M$, we find

$$\ell_{\text{SYM}} \sim \frac{1}{\Lambda_{\text{unbroken}}} \sim \frac{1}{m_{\text{4d}}^{F/3N} \Lambda^{1-F/3N}} \, . \tag{5.8}$$

We see that in the $m_{\text{4d}} \to 0$ limit, the thickness of the SYM transition is parametrically smaller than the size of the full domain wall. We conclude that, in that limit, the SYM transition can be treated as sharp, or "instantaneous". Thus, we can construct domain walls in which we abruptly jump from one sheet to another at points along the path. The worldvolume theory on one such domain wall (besides the center of mass) consists of the AV topological sector associated to the jump times possible Goldstone fields for broken symmetries. More specifically, for each one of such jumps the worldvolume theory acquires a CS topological sector

$$U(\Delta)_{N-F-\Delta, N-F} \tag{5.9}$$

(using $\mathcal{N} = 0$ notation), whenever $e^{2\pi i \Delta/(N-F)}$ is the phase shift in the $SU(N-F)$ sector.

When we jump from one sheet to another, the value of $W$ changes (at fixed $M$). Each smooth portion of the profile, satisfying the differential equation (5.1), must map to a straight line with direction $e^{i\gamma}$ on the complex $W$-plane, where $\gamma$ is the phase of the central charge (2.3) —and similarly each jump due to a SYM wall must point in the same direction of $e^{i\gamma}$—because the preserved supercharges are constant throughout the wall. This implies that each smooth portion is in the pre-image of a segment along the straight line connecting $W_{-\infty}$ to $W_{+\infty}$. If we draw on the $M$-plane all pre-images of the straight line, a domain wall will be given by a continuous, piecewise $C^{\infty}$ path along those pre-images, from one vacuum to another. This procedure will become clearer in the examples we will discuss next.

To sum up, we can divide the various domain walls at $m_{\text{4d}} \ll \Lambda$ into two groups. The first group consists of symmetry preserving walls, that can be studied algebraically. The associated three-dimensional vacuum is gapped, either trivially (for standard WZ walls) or hosting a topological sector (for hybrid walls, whenever the path on the mesonic space undergoes one or more jumps in the unbroken $SU(N-F)$ SYM). The second group consists of symmetry breaking walls, and it requires the solution of ODEs. The three-dimensional vacuum accommodates a supersymmetric NLSM of Goldstone fields. This can be accompanied, again, by a non-trivial topological theory (for hybrid walls) if a jump in the underlying $SU(N-F)$ SYM occurs. In the following, we will discuss symmetry preserving and symmetry breaking domain walls in turn.

In Table 2 we list all BPS domain walls of $SU(N)$ SQCD with $F < N$, up to rank $N = 5$, in the regime of small mass $m_{\text{4d}} \ll \Lambda$, as predicted by the worldvolume analysis of Section 4 and already packaged in Table 1. We only indicate $k$-walls with $1 \le k \le N/2$, since the remaining ones are obtained by applying a parity transformation to $(N-k)$-walls. Our goal is to reproduce all such domain walls by the aforementioned 4d analysis.

Let us stress that, for fixed soliton sector $k$, namely for fixed 4d vacua on the left and on the right, we find in general more than one BPS wall. In the three-dimensional worldvolume description, they correspond to different vacua labelled by $J$. Such walls are physically inequivalent, not related by any symmetry, and yet they are exactly degenerate in tension. This, of course, is an effect of bulk supersymmetry which fixes the tension in terms of $|\Delta W|$.

## 5.1 Symmetry preserving walls

In this section we restrict to domain walls that do not break the $SU(F)$ mesonic symmetry, in other words we take $M = \widetilde{M} \mathbb{1}_F$ all along the domain wall trajectory. Notice that this is automatically the case when $F = 1$. For a WZ theory of a single chiral superfield, the domain wall equation is an ODE of a single variable. If we are only interested in the orbit of the field,

Table 2: List of all BPS domain walls of massive $SU(N)$ SQCD with $F < N$, for $m_{4d} \ll \Lambda$, up to rank $N = 5$ (extracted from Table 1). We only indicate $k$-walls with $k \leq N/2$, since the remaining ones with $N/2 < k < N$ are obtained applying a parity transformation. In each soliton sector $k$, we list the worldvolume theories on different domain walls (for topological sectors we use here the $\mathcal{N} = 0$ notation). Trivially gapped vacua are indicated by "gap".

| $SU(2)$ | $F = 1$ | $k = 1$ : gap, gap |
|---|---|---|

| $SU(3)$ | $F = 2$ | $k = 1$ : gap, $\mathbb{P}^1$ |
|---|---|---|
| | $F = 1$ | $k = 1$ : $U(1)_2$, gap |

| $SU(4)$ | $F = 3$ | $k = 1$ : gap, $\mathbb{P}^2$ | $k = 2$ : $\mathbb{P}^2$, $\mathbb{P}^2$ |
|---|---|---|---|
| | $F = 2$ | $k = 1$ : $U(1)_2$, $\mathbb{P}^1$ | $k = 2$ : gap, $U(1)_2 \times \mathbb{P}^1$, gap |
| | $F = 1$ | $k = 1$ : $U(1)_3$, gap | $k = 2$ : $U(1)_{-3}$, $U(1)_3$ |

| $SU(5)$ | $F = 4$ | $k = 1$ : gap, $\mathbb{P}^3$ | $k = 2$ : $\mathbb{P}^3$, $\mathrm{Gr}(2,4)$ |
|---|---|---|---|
| | $F = 3$ | $k = 1$ : $U(1)_2$, $\mathbb{P}^2$ | $k = 2$ : gap, $U(1)_2 \times \mathbb{P}^2$, $\mathbb{P}^2$ |
| | $F = 2$ | $k = 1$ : $U(1)_3$, $\mathbb{P}^1$ | $k = 2$ : $U(1)_{-3}$, $U(1)_3 \times \mathbb{P}^1$, gap |
| | $F = 1$ | $k = 1$ : $U(1)_4$, gap | $k = 2$ : $U(2)_{2,4}$, $U(1)_4$ |

*i.e.* on the image of the field in the complex $\widetilde{M}$-plane, and not in the precise profile as a function of $x$, then the problem becomes algebraic: we only need to invert the function $W(\widetilde{M})$. This is equivalent to the fact that $\mathrm{Im}\left(e^{-i\gamma} W\right)$ is constant through the wall. Applying the algebraic method we will be able to determine all domain walls—and be sure we are not missing any.

It is convenient to express $M$ in units of $\left(\Lambda^{3N-F}/m_{4d}^{N-F}\right)^{1/N}$ to make it dimensionless and so that the vacua lie on the unit circle. This operation rescales the Kähler potential as well. In these units the superpotential (3.1) and vacua (3.2) become

$$W = \Lambda^{\frac{3N-F}{N}} m_{4d}^{\frac{F}{N}} \left[ \mathrm{Tr}\, M + (N-F)\left(\frac{1}{\det M}\right)^{\frac{1}{N-F}} \right], \qquad M = e^{\frac{2\pi i}{N} k} \mathbb{1}_F . \tag{5.10}$$

Setting to one the remaining dimensionfull constant $\Lambda^{\frac{3N-F}{N}} m_{4d}^{\frac{F}{N}}$, the restriction of the superpotential (5.10) to the symmetry-preserving slice is

$$W = F \widetilde{M} + \frac{N-F}{\left(\widetilde{M}^F\right)^{\frac{1}{N-F}}} . \tag{5.11}$$

This is a multi-valued function of $\widetilde{M}$ with $N-F$ sheets above each point (corresponding to the different vacua of the unbroken $SU(N-F)$ SYM theory). It turns out that the sheets arrange into $d = \gcd(N, F)$ disconnected components.[19] For convenience, we can introduce a covering variable $X$ such that

$$\widetilde{M} = X^{(N-F)/d} , \tag{5.12}$$

---

[19]These components only touch at the origin, which however is a singular point and should be excised. We stress that the covering space splits into disconnected components only after restricting to the symmetry-preserving slice.

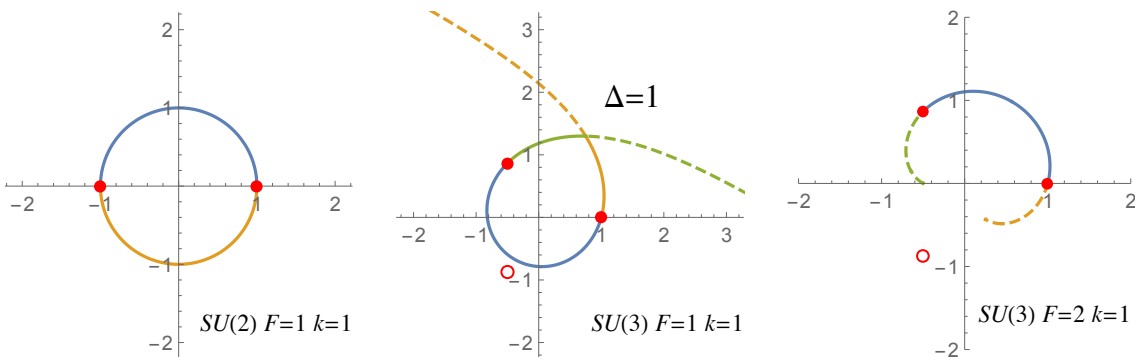

Figure 2: Symmetry preserving $k$-walls in massive $SU(N)$ SQCD with $F < N$ flavors for $N = 2, 3$. Each figure refers to fixed values of $N, F$ and soliton sector $k$. We draw the $N$ vacua (red dots and circles) on the $\widetilde{M}$-plane, as well as the pre-image of a straight line in the $W$-plane connecting $W_{-\infty}$ to $W_{+\infty}$. The pre-image consists of $N$ curves, drawn in different colors. Paths connecting the vacua and thus corresponding to domain walls are solid lines, while the rest of the pre-image is dashed. If a path involves a jump from one sheet to another (namely, from one portion of the pre-image to another), we indicate the value of $\Delta$.

and write

$$W_{(a)} = F X^{(N-F)/d} + e^{\frac{2\pi i}{N-F} a} \frac{N-F}{X^{F/d}} \, , \qquad a = 0, \ldots, d-1 \, . \tag{5.13}$$

Each branch $W_{(a)}$ is a single-valued function on its domain, which covers the $\widetilde{M}$-plane $\frac{N-F}{d}$ times, and there are $d$ disconnected domains labelled by $a$. Each domain hosts $N/d$ vacua, located at

$$X = \exp\left\{ \frac{2\pi i}{N} \left( \frac{d\,a}{N-F} + d\,j \right) \right\} \qquad i.e. \qquad \widetilde{M} = \exp\left\{ \frac{2\pi i}{N} \big( a + (N-F)\,j \big) \right\} \, , \tag{5.14}$$

where $j = 0, \ldots, \frac{N}{d} - 1$. At the vacua, $W_{(a)} = N\,\widetilde{M}$.

Without loss of generality, we choose the vacuum $\widetilde{M} = 1$ at $x = -\infty$, and the vacuum $\widetilde{M} = e^{\frac{2\pi i}{N} k}$ at $x = +\infty$. We ask if they can be connected by $k$-walls, whose central charge would be

$$Z = 2 \Delta W = 2N \big( e^{\frac{2\pi i}{N} k} - 1 \big) = e^{i\gamma} |Z| \, . \tag{5.15}$$

We connect the corresponding points on the $W$-plane by a straight line (with direction $e^{i\gamma}$), and compute its pre-image (consisting of $N$ parts) on the full domain. If there exists a continuous curve on the covering space $X$ connecting the two vacua, this is a standard WZ wall. Its worldvolume theory is trivially gapped (besides the free center of mass) because no continuous symmetry gets broken. On the contrary, there can exist curves that are continuous on the $\widetilde{M}$-plane but include jumps on the covering spaces $X_{(a)}$—either within the same domain or from one domain to another. These are walls that combine the WZ evolution with sharp (in the $m_{4d} \to 0$ limit) AV walls in the unbroken $SU(N-F)$ gauge theory, as previosuly discussed. For each jump $\Delta$, the worldvolume theory acquires a topological sector $U(\Delta)_{N-F-\Delta, N-F}$. As we will see in the examples below, we observe that walls involve at most one jump.

### 5.1.1 Examples

Consider first $SU(2)$ SQCD with $F = 1$. The theory has two vacua, and so there is only one possible soliton sector, $k = 1$. Since $F = 1$, the meson field has only one component, all walls can be found algebraically and none of them can break the flavor symmetry. Morever, since

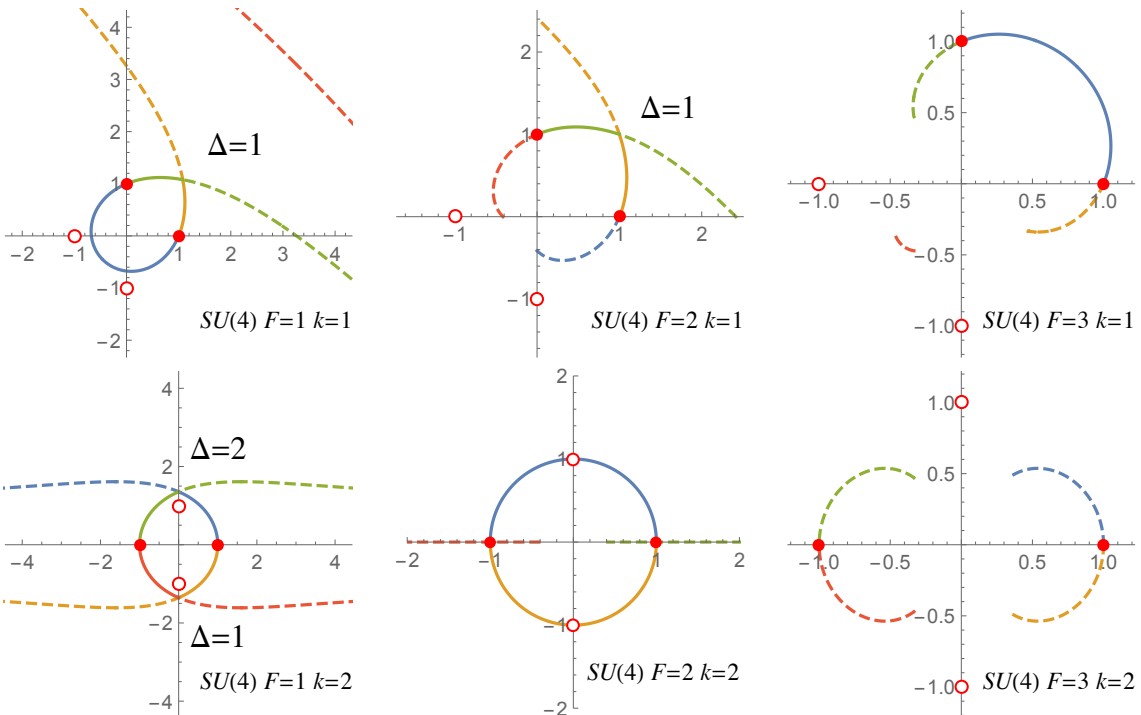

Figure 3: Symmetry preserving $k$-walls in massive $SU(4)$ SQCD with $F = 1, 2, 3$ flavors. Conventions are the same as in Figure 2.

$F = N - 1$, there is no unbroken gauge sector on the mesonic space, the superpotential is single-valued, all walls are visible in the WZ description and their worldvolume theory cannot host any topological sector. As shown in Figure 2 left, the pre-image of a straight line on the $W$-plane gives two domain walls (blue and yellow in the figure) whose worldvolume theory is trivially gapped. This agrees with Table 2.

Consider now $SU(3)$ SQCD. The theory has three vacua, so there are two soliton sectors, $k = 1, 2$. However, the sector $k = 2$ is the parity reversal of the sector $k = 1$ and thus we only study the latter. For $F = 1$ (Figure 2 center) all domain walls can be found algebraically. We find a WZ wall (blue in the figure) whose worldvolume theory is trivially gapped. We also find a wall that involves the jump from one sheet of $W$ to the other (yellow followed by green in the figure). This corresponds to a jump of vacuum (indicated as $\Delta = 1$) in the unbroken $SU(2)$ gauge theory, giving rise to the topological theory $U(1)_2$. For $F = 2$ (Figure 2 right) all domain walls are of WZ type. Restricting to symmetry preserving walls, we find one (blue in the figure) whose worldvolume theory is trivially gapped. This matches, again, with Table 2, as far as symmetry preserving walls are concerned.

Consider then $SU(4)$ SQCD. This theory has four vacua and we study the soliton sectors $k = 1, 2$ (the sector $k = 3$ is the parity reversal of the sector $k = 1$). For $F = 1$ (Figure 3 left) all domain walls can be found algebraically. In the sector $k = 1$ we find a trivially gapped wall (blue) and a wall with $U(1)_3$ topological sector (yellow followed by green) from the $\Delta = 1$ jump in the unbroken $SU(3)$. In the sector $k = 2$ we find a wall with topological sector $U(2)_{1,3} \cong U(1)_{-3}$ (blue followed by green) from a $\Delta = 2$ jump in the unbroken $SU(3)$, and a wall with topological sector $U(1)_3$ (yellow followed by red). For $F = 2$ (Figure 3 center) the domain of the restriction of the superpotential to symmetry preserving configurations has two disconnected components. In the sector $k = 1$ the two vacua live on disconnected domains, and symmetry preserving walls must necessarily involve a jump from one sheet to the other. Indeed, we find one such wall (yellow followed by green) hosting a topological sector $U(1)_2$.

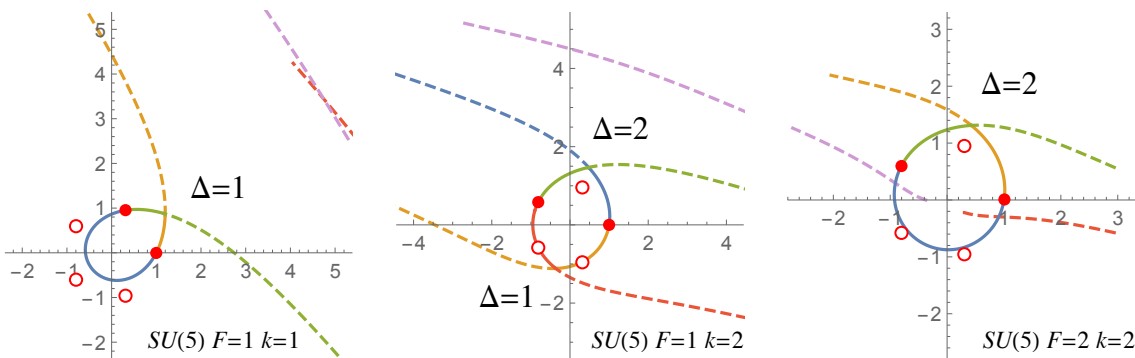

Figure 4: Symmetry preserving $k$-walls in massive $SU(5)$ SQCD with $F$ flavors, for some selected values of $F$ and soliton sector $k$. Conventions are again as in Figure 2.

In the sector $k = 2$ the two vacua live on the same domain, and we find two WZ walls (blue and yellow) with trivially gapped worldvolume theory. Finally, for $F = 3$ (Figure 3 right) all domain walls are of WZ type. Restricting to the symmetry preserving ones, we find one (blue) in the sector $k = 1$, and none in the sector $k = 2$. All these results match with Table 2.

As last example, consider $SU(5)$ SQCD. The independent soliton sectors are $k = 1, 2$, while $k = 3, 4$ are their parity reversal. We report our results in some selected cases, only. For $F = 1$, in the sector $k = 1$ (Figure 4 left) we find a WZ wall (blue) with trivially gapped vacuum, and a wall (yellow followed by green) with topological sector $U(1)_4$. In the sector $k = 2$ (Figure 4 center) we find a wall (blue followed by green) with topological sector $U(2)_{2,4}$ and a wall (yellow followed by red) with topological sector $U(1)_4$. For $F = 2$, in the sector $k = 2$ (Figure 4 right) we find a WZ wall (blue) with trivially gapped vacuum, and a wall (yellow followed by green) with topological sector $U(2)_{1,3} \cong U(1)_{-3}$. We find again full agreement with Table 2.

## 5.2 Symmetry breaking walls

Let us now discuss the more general type of domain walls, those through which the meson field $M$ is not proportional to the identity, despite being proportional to the identity in the vacua on the two sides. With more than one independent component, we have no options but directly solve the differential equations (5.1).

Let us assume that, at least at one point along the domain wall profile, the meson matrix is diagonalizable. We can use an $SU(F)$ flavor rotation to bring $M$ to a diagonal form at that point. Then, the ODEs (5.1) imply that $M(x)$ remains diagonal for all values of the spatial coordinate $x$. Indeed, the Kähler metric that follows from the Kähler potential (5.5), evaluated at points where $M_{ij} = \lambda_j \delta_{ij}$ is a diagonal matrix and where $\overline{M} = M^\dagger$, takes the form

$$\frac{\partial^2 \mathcal{K}}{\partial M_{ij} \partial \overline{M}_{ba}}\bigg|_{\text{diag}} = \frac{\delta_{jb}\,\delta_{ai}}{|\lambda_i| + |\lambda_j|}\,. \tag{5.16}$$

Since at these points also the gradient of the superpotential is a diagonal matrix, it follows from eqn. (5.1) that the spatial derivative of $M(x)$ is diagonal as well. We can thus restrict to ODEs for the diagonal components $\lambda_j(x)$.

As before, it is convenient to rescale the meson field $M$ and the spatial coordinate $x$ (as

well as to possibly shift the phase of the central charge)[20] as

$$M \rightarrow \left(\frac{\Lambda^{3N-F}}{m_{\mathrm{4d}}^{N-F}}\right)^{\frac{1}{N}} M\,, \qquad\qquad x \rightarrow \frac{x}{|m_{\mathrm{4d}}|}\,, \tag{5.17}$$

in order to obtain dimensionless differential equations,

$$\partial_x \lambda_j = 2e^{i\gamma}|\lambda_j|\left(1 - \frac{1}{\lambda_j^*\left(\prod_k \lambda_k^*\right)^{1/(N-F)}}\right) \qquad \text{for } j = 1,\ldots,F\,. \tag{5.18}$$

We decompose the eigenvalues $\lambda_j$ into radial and polar parts, $\lambda_j = \rho_j e^{i\phi_j}$. This gives the system of differential equations

$$\partial_x \rho_j = 2\rho_j \cos(\gamma - \phi_j) - \frac{2\cos\left(\gamma + \frac{1}{N-F}\sum_k \phi_k\right)}{\prod_k \rho_k^{1/(N-F)}}\,,$$

$$\partial_x \phi_j = 2\sin(\gamma - \phi_j) - \frac{2\sin\left(\gamma + \frac{1}{N-F}\sum_k \phi_k\right)}{\rho_j \prod_k \rho_k^{1/(N-F)}}\,. \tag{5.19}$$

This system is of Hamiltonian type: $\partial_x \rho_j = \partial H/\partial \phi_j$ and $\partial_x \phi_j = -\partial H/\partial \rho_j$ with

$$H = -2\sum_k \rho_k \sin(\gamma - \phi_k) - \frac{2(N-F)\sin\left(\gamma + \frac{1}{N-F}\sum_k \phi_k\right)}{\prod_k \rho_k^{1/(N-F)}} = 2\,\mathbb{Im}\left(e^{-i\gamma}W\right). \tag{5.20}$$

Consistently, $\mathbb{Im}\left(e^{-i\gamma}W\right)$ is a "constant of motion" along the domain wall profile.

Let us recall that the effective superpotential on the mesonic space is multi-valued, due to the unbroken $SU(N-F)$ SYM theory at generic points. We can work in a connected covering space, with covering order $N-F$, defined by

$$\phi_1 \cong \phi_1 + 2\pi(N-F)\,, \qquad\qquad \phi_i, \phi_j \cong \phi_i + 2\pi, \phi_j - 2\pi\,. \tag{5.21}$$

On the covering space, the $N$ vacua are located at

$$\rho_1 = \ldots = \rho_F = 1\,, \qquad \phi_1 = \ldots = \phi_{F-1} = \frac{2\pi}{N}k\,, \qquad \phi_F = \frac{2\pi}{N}k - 2\pi k\,, \tag{5.22}$$

and are labelled by $k = 0,\ldots,N-1$.

Domain walls of the standard WZ type correspond to solutions to eqns. (5.19) that are continuous on the covering space. The worldvolume theory on such domain walls does not include any topological sector. For more general hybrid domain walls, at certain spatial locations $x_*$ the profile jumps from one sheet of the covering to another (the $\lambda_j$'s remain continuous). This corresponds to a shift

$$\phi_1 \rightarrow \phi_1 - 2\pi\Delta\,, \tag{5.23}$$

i.e. a shift of vacuum in the unbroken $SU(N-F)$ SYM, and the worldvolume theory includes a topological sector $U(\Delta)_{N-F-\Delta,N-F}$. A non-trivial prediction of the 3d analysis is that, even for symmetry breaking walls, solutions can accommodate at most one jump, as we found for symmetry preserving walls.

---

[20]Precisely, if $\Lambda$ and $m_{\mathrm{4d}}$ are not real positive, we should rescale $e^{i\gamma}x \rightarrow \mathrm{phase}\left(\Lambda^{3-F/N}m_{\mathrm{4d}}^{F/N}\right)e^{i\gamma}x/|m_{\mathrm{4d}}|$. From the rescaling of the spatial coordinate needed to make the ODEs dimensionless one can extract the typical size of SQCD domain walls, as in (5.7).

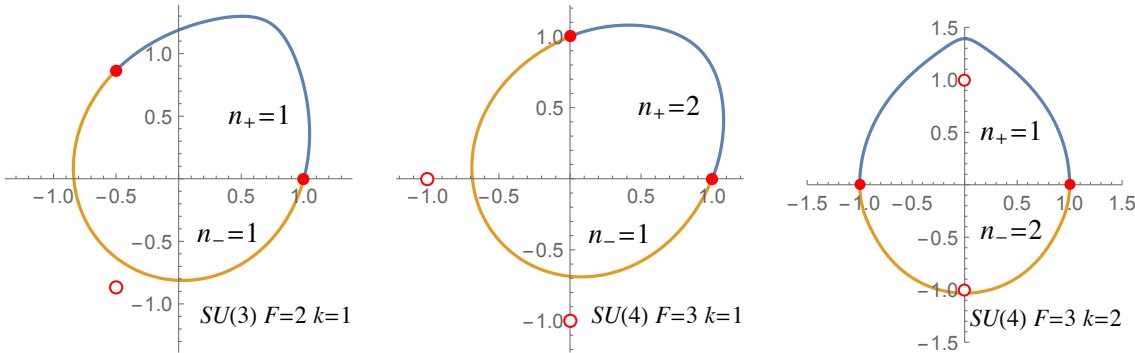

Figure 5: Symmetry breaking $k$-walls in massive $SU(N)$ SQCD with $N = 3, 4$ and $F = N - 1$ flavors. Each figure refers to a specific domain wall for given values of $N, F, k, J$. We draw the (smooth) orbits in the complex plane of a solution to the differential equations (5.19), in which $n_+$ eigenvalues are equal to $\lambda_+$ and $n_-$ are equal to $\lambda_-$ ($n_+ + n_- = F$).

When the meson field $M$ is not proportional to the identity matrix along the profile, namely when the eigenvalues $\lambda_j$ are not all equal, the domain wall spontaneously breaks the flavor symmetry $SU(F)$. Another non-trivial prediction of the analysis of Section 4 is that the eigenvalues split at most into two groups. Calling $n_\pm$ the number of eigenvalues in the first and second group, respectively, with $n_+ + n_- = F$, the worldvolume theory on the domain wall hence includes an

$$\mathcal{N} = 1 \quad \text{NLSM} \qquad \frac{U(F)}{U(n_+) \times U(n_-)} \, . \tag{5.24}$$

It would be nice to understand analytically why there cannot be solutions to (5.19) in which the eigenvalues organize into three or more distinct groups, or which undergo two or more jumps on the covering space.

### 5.2.1 Examples

In Section 5.1.1 we were able to determine algebraically the full set of symmetry preserving domain wall solutions for the cases considered. For symmetry breaking walls we need to solve ODEs. This can be done numerically using a shooting technique. We will be able to explicitly construct all domain wall solutions predicted by the 3d analysis of Section 4 and summarized in Table 2 for low ranks. However, we will not be able to prove that no other domain wall solutions can exist.

Without loss of generality, a $k$-wall connects the vacuum at $\{\lambda_j = 1\}$ to the vacuum at $\{\lambda_j = e^{2\pi i k/N}\}$. To construct numerical solutions it is convenient to set the origin $x = 0$ in the middle of the wall. We divide the eigenvalues into two groups $\lambda_\pm$ of $n_\pm$ elements, respectively. By reflection symmetry with respect to the origin, we set the phases of the eigenvalues equal to $\pm e^{\pi i k/N}$ at $x = 0$. The known value of the constant of motion $H$ enforces a relation between $\rho_+(0)$ and $\rho_-(0)$. This leaves us with a shooting problem with one initial condition at $x = 0$, to be found such that the eigenvalue profiles hit the vacua at $x = \pm\infty$. For domain walls with no jump, symmetry guarantees that a solution that hits the vacuum at $x = -\infty$ also hits the vacuum at $x = +\infty$. For domain walls with a jump at $x = 0$, instead, we solve the shooting problem on the half-line $x > 0$, then the jump must be such that the solution automatically hits the other vacuum at $x = -\infty$.

Consider first $SU(3)$ SQCD with $F = 2$. Table 2 predicts a symmetry breaking domain wall with $n_+ = n_- = 1$ in the $k = 1$ soliton sector. We draw the corresponding numerical solution

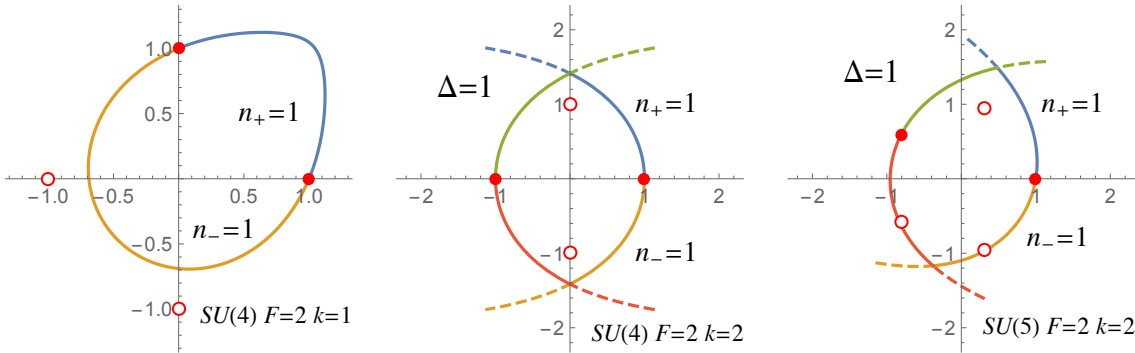

Figure 6: Symmetry breaking $k$-walls for $N = 4, 5$ and $F = 2$ flavors. The central and right figures contain a continuous but not smooth profile, that involves a jump from one sheet to another (the value of $\Delta$ is indicated). Solid curves represent the orbits followed by the eigenvalues $\lambda_{\pm}$, while dashed curves are their smooth continuation as solutions to (5.19).

in Figure 5 left, in which the orbits of the two eigenvalues $\lambda_{\pm}$ on the complex plane are in blue and yellow, respectively.

Consider now $SU(4)$ SQCD. For $F = 3$, the superpotential is a single-valued function and domain walls do not involve jumps. As predicted by Table 2, in the $k = 1$ soliton sector we find one domain wall with $n_+ = 2$, $n_- = 1$ (Figure 5 center). In the $k = 2$ soliton sector we find two domain walls with $n_+ = 2$, $n_- = 1$: one (Figure 5 right) is the complex conjugate of the other. For $F = 2$ there is an unbroken $SU(2)$ gauge theory on the mesonic space, the superpotential is double-valued and jumps are possible. In the $k = 1$ soliton sector we find a continuous domain wall with $n_+ = n_- = 1$ (Figure 6 left). In the $k = 2$ soliton sector, instead, we find a domain wall with $n_+ = n_- = 1$ that involves a $\Delta = 1$ jump (Figure 6 center): one eigenvalue draws the blue followed by green orbit, while the other one draws the yellow followed by red orbit. The wordvolume theory is thus a $\mathbb{P}^1$ NLSM times a $U(1)_2$ topological sector, as predicted again in Table 2.

Symmetry breaking domain walls of SQCD with higher gauge rank can be studied similarly, finding perfect agreement with Table 2. As a selected example, $SU(5)$ SQCD with $F = 2$, in the $k = 2$ soliton sector has a domain wall with $n_+ = n_- = 1$ and a $\Delta = 1$ jump in the unbroken $SU(3)$ gauge theory (Figure 6 right). Its worldvolume theory is thus $U(1)_3 \times \mathbb{P}^1$.

# 6 Domain walls of $Sp(N)$ SQCD

In this section we extend the previous discussions to SQCD with symplectic gauge group. As we will see, the story is very similar to the $SU(N)$ case.

Let us consider four-dimensional $\mathcal{N} = 1$ $Sp(N) \equiv USp(2N)$ SQCD with $F < N + 1$ flavors, namely with $2F$ chiral superfields $Q_i$ in the fundamental representation, where the flavor index is $i = 1, \ldots, 2F$. The gauge group $Sp(N)$ is the subgroup of $SU(2N)$ that leaves the $2N \times 2N$ symplectic form $\Omega = \mathbb{1}_N \otimes i\sigma_2$ invariant and its dimension is $N(2N + 1)$.[21] In the massless theory, the continuous non-anomalous global symmetry is $SU(2F) \times U(1)_R$.

Very much like $SU(N)$ SQCD with $F < N$ flavors, in $Sp(N)$ SQCD a non-perturbative run-

---

[21]In our conventions $Sp(1) \equiv USp(2) \cong SU(2)$.

away effective superpotential on the mesonic space is generated if $F < N + 1$ [23, 76]:

$$W_{\text{ADS}} = (N + 1 - F)\left(\frac{\Lambda^{3(N+1)-F}}{\text{Pf}\,M}\right)^{\frac{1}{N+1-F}}, \tag{6.1}$$

where $M_{ij} = \Omega_{\alpha\beta} Q_i^{\alpha} Q_j^{\beta}$ is the anti-symmetric $2F \times 2F$ mesonic matrix and Pf stands for Pfaffian.[22] As before, we turn on a diagonal mass term for the flavors:

$$W_m = \frac{m_{\text{4d}}}{2} M_{ij} \Omega^{ij}, \tag{6.2}$$

where $\Omega^{ij}$ is the symplectic form of $Sp(F)$ with $i, j = 1, \ldots, 2F$ (in the following, we will indicate all symplectic forms as $\Omega$, irrespective of their dimension, and will not distinguish between upper and lower indices). The mass term stabilizes the runaway directions. It also explicitly breaks the $SU(2F)$ flavor symmetry to $Sp(F)$, while leaving a discrete $\mathbb{Z}_{2(N+1)}$ R-symmetry unbroken.[23] The mesons transform in the rank-two antisymmetric representation of $Sp(F)$. The full effective superpotential on the mesonic space reads

$$W = \frac{m_{\text{4d}}}{2} M_{ij} \Omega^{ij} + (N + 1 - F)\left(\frac{\Lambda^{3(N+1)-F}}{\text{Pf}\,M}\right)^{\frac{1}{N+1-F}}. \tag{6.3}$$

The theory develops gaugino condensation giving rise to $N + 1$ gapped vacua, corresponding to the spontaneous R-symmetry breaking $\mathbb{Z}_{2(N+1)} \to \mathbb{Z}_2$. The $N + 1$ vacua are rotated into each other by the broken generators and sit on the mesonic space at

$$M = \widetilde{M}\,\Omega_{2F}, \qquad\qquad \widetilde{M}^{N+1} = \frac{\Lambda^{3(N+1)-F}}{m_{\text{4d}}^{N+1-F}}. \tag{6.4}$$

We want to study domain walls interpolating between these vacua.

**Domain wall trajectories.** The mathematical problem of studying domain wall solutions for $Sp(N)$ SQCD with $F < N + 1$ flavors is equivalent to the one for $SU(N + 1)$ SQCD with $F < N + 1$ flavors. Indeed, upon diagonalizing the $2F \times 2F$ antisymmetric mesonic matrix and rescaling to dimensionless quantities, eqn. (6.3) becomes

$$W = \sum_{j=1}^{F} \lambda_j + (N + 1 - F)\prod_{j=1}^{F} \lambda_j^{-\frac{1}{N+1-F}}, \tag{6.5}$$

which is the same as the effective superpotential we discussed in Section 5, upon shifting $N \to N + 1$ everywhere there. In other words, the ODEs that determine the domain wall trajectories are the same for $SU(N + 1)$ and $Sp(N)$ gauge groups. Hence, for small values of the flavor masses, $m_{\text{4d}} \ll \Lambda$, we obtain the same structure of multiple vacua corresponding to different classes of domain walls, preserving or partially breaking the $Sp(F)$ flavor symmetry, and with or without a topological sector. For large flavor masses, instead, the domain wall theory should reduce to that of pure $Sp(N)$ SYM. Finally, at some value $m_{\text{4d}}^*$ of the 4d mass (that could depend on $N, F, k$, and that corresponds to $m = 0$ in the three-dimensional field theory description), a single second-order phase transition should occur, where multiple vacua coalesce.

---

[22]The Pfaffian of a $2F \times 2F$ antisymmetric matrix $M$ is $\text{Pf}\,M = \frac{1}{2^F F!} M_{i_1 i_2} \ldots M_{i_{2F-1} i_{2F}} \epsilon^{i_1 \cdots i_{2F}}$ and it satisfies $(\text{Pf}\,M)^2 = \det M$. Its variation is $\delta\,\text{Pf}\,M = \frac{1}{2}\text{Pf}\,M \cdot \text{Tr}(M^{-1}\delta M)$. Moreover $\text{Pf}\,\Omega = 1$.

[23]Notice that this R-symmetry gives charge 1 to the flavor fields $Q$. In particular, it is not a subgroup of the continuous $U(1)_R$ R-symmetry of the massless theory.

In the following, we present our proposal for the 3d theory living on $k$-walls of $Sp(N)$ SQCD with $F < N + 1$ flavors, and show that the domain walls we find are in one-to-one correspondence with those of $SU(N + 1)$ SQCD with $F < N + 1$ flavors. The difference is that the TQFTs are CS theories with $Sp(k)$ gauge group instead of $U(k)$, and the supersymmetric NLSMs have target spaces given by the quaternionic Grassmannians

$$\mathrm{HGr}(J, F) = \frac{Sp(F)}{Sp(J) \times Sp(F - J)} = \mathrm{HGr}(F - J, F) \tag{6.6}$$

instead of $\mathrm{Gr}(J, F)$.

## 6.1 Three-dimensional worldvolume theory

The 3d theory we propose to describe the $k$-walls of $Sp(N)$ SQCD with $F$ flavors (for $0 < k < N+1$ and $F < N + 1$) is

$$
\begin{aligned}
&\text{3d } \mathcal{N} = 1 \; Sp(k)_{N+1-\frac{F}{2}} \text{ gauge theory} \\
&\text{with a rank-2 antisymmetric scalar multiplet } \Phi \\
&\text{and } F \text{ fundamental scalar multiplets } X \, ,
\end{aligned}
\tag{6.7}
$$

and no bare superpotential involving $\Phi$. We indicate the fundamentals by the matrix $X_{ai}$ where $a = 1, \ldots, 2k$ is the gauge index and $i = 1, \ldots, F$ is the flavor index. As usual when dealing with pseudo-real representations, it is convenient to double the number of fundamentals: we introduce $X_{aI}$ taking $I = 1, \ldots, 2F$ and then impose the reality condition $X_{aI} = \Omega^{ab} \Omega^{IJ} X_{bJ}^*$. This makes manifest the $Sp(F)$ flavor symmetry that acts on the $F$ fundamentals. Gauge invariants are constructed in terms of

$$X_{IJ}^2 \equiv X_{aI} X_{bJ} \Omega^{ab} = X_{aI} X_{aK}^* \Omega^{KJ} \, , \tag{6.8}$$

which are antisymmetric in $IJ$.

The representation of $\Phi$ breaks into two irreducible representations: a singlet (proportional to $\Omega$), which is the Goldstone mode associated to broken translations, and the $\Omega$-traceless antisymmetric representation which classically gives rise to flat directions. Quantum corrections lift those flat directions, generating a negative mass around $\Phi = 0$. Integrating out the traceless antisymmetric (whose quadratic Casimir is $k-1$) we obtain the simpler low-energy description

$$\text{3d } \mathcal{N} = 1 \; Sp(k)_{N+1-\frac{F+k-1}{2}} \text{ with } F \text{ flavors } X \tag{6.9}$$

with superpotential

$$\mathcal{W} = \frac{1}{16} \mathrm{Tr} X^2 \Omega X^2 \Omega + \frac{\alpha}{16} \big(\mathrm{Tr} X^2 \Omega\big)^2 - \frac{m}{4} \mathrm{Tr} X^2 \Omega \, . \tag{6.10}$$

Notice that $-\frac{m}{4} \mathrm{Tr} X^2 \Omega = \frac{m}{2} \sum_{ai} X_{ai} X_{ai}^*$. We assume $\alpha > -\min(k, F)^{-1}$. Before discussing its vacuum structure, let us notice that our proposal already passes a non-trivial check. As we show below, this theory enjoys a single gapped vacuum for $m > 0$ and multiple vacua for $m < 0$, with a second-order phase transition at $m = 0$, very much like in $SU(N)$ SQCD. Due to the broken R-symmetry, $k$-walls are the parity reversal of $(N + 1 - k)$-walls. Hence, according to (6.9), this should imply the following 3d $\mathcal{N} = 1$ duality to hold at the phase transition:

$$
\begin{array}{ccc}
\mathcal{N} = 1 \;\; Sp(k)_{N - \frac{F+k-3}{2}} & \quad & \mathcal{N} = 1 \;\; Sp(N + 1 - k)_{-1 - \frac{N+k-F}{2}} \\
\text{with } F \text{ flavors and quartic } \mathcal{W} & \longleftrightarrow & \text{with } F \text{ flavors and quartic } \mathcal{W} \, .
\end{array}
\tag{6.11}
$$

The sign of the quartic couplings is equal to the sign of the CS level. This is indeed one of the 3d dualities recently proposed in [18] and expected to be valid precisely in the regime of interest, *i.e.* $0 < k < N + 1$. Notice that for $k = 1$, the dual description is a three-dimensional $\mathcal{N} = 1$ CS theory $Sp(N)_{-1-\frac{N+1}{2}+\frac{F}{2}}$ with $F$ flavors: intriguingly, the gauge group is the same as the 4d one, suggesting a possible connection with an interface operator.

In the following we will provide further checks of this duality, showing that as the mass parameter $m$ is turned on and varied from positive to negative values, the vacuum structure of the theory on the left-hand-side of (6.11) is the same as that of the theory on the right. Notice that a mass term $-\text{Tr} X^2 \Omega$ on the left is mapped to a term $\text{Tr} Y^2 \Omega$ on the right. We will sometimes call theory A the theory on the left-hand-side of (6.11) and theory B the one on the right. Since most of the logic is the same as in Section 4, in what follows we will skip all unnecessary details.

Let us now discuss the vacuum structure of the theory.

• **$m > 0$.** It is not difficult to see that, in this regime, for both theories A and B there exists a unique vacuum, and the duality (6.11) reduces to the 3d $\mathcal{N} = 1$ duality

$$\mathcal{N} = 1 \quad Sp(k)_{N-\frac{k-3}{2}} \qquad \longleftrightarrow \qquad \mathcal{N} = 1 \quad Sp(N+1-k)_{-1-\frac{N+k}{2}} \, . \tag{6.12}$$

This duality is known to hold since, upon integrating out the massive gaugini, it boils down to the level/rank duality [8]

$$Sp(k)_{N+1-k} \qquad \longleftrightarrow \qquad Sp(N+1-k)_{-k} \, , \tag{6.13}$$

valid for $0 \le k \le N + 1$. This provides a simple check of the duality (6.11).

**Domain walls of $Sp(N)$ SYM.** As a consequence of our proposal, we find that the $\mathcal{N} = 1$ theory on the left-hand-side of (6.12) describes a $k$-wall of 4d $\mathcal{N} = 1$ pure $Sp(N)$ SYM. The duality (6.12) represents the fact that a $k$-wall is the parity reversal of an $(N+1-k)$-wall. Reinstating the massive scalar multiplet $\Phi$ that describes the center-of-mass motion as well as the breaking of a $k$-wall into $k$ 1-walls, we have

$$\text{3d } \mathcal{N} = 1 \ Sp(k)_{N+1} \text{ gauge theory with a (rank-2 antisymmetric) scalar multiplet } \Phi \, . \tag{6.14}$$

These are the natural generalizations of the Acharya-Vafa domain wall theories to the case of four-dimensional $\mathcal{N} = 1 \ Sp(N)$ SYM.

• **$m < 0$.** In this regime we get vacua where $J$ flavors take a VEV, with $J \le \min(k, F)$. In order to avoid confusion, for theory B we parameterize the vacua with the integer $H \le \min(N-k+1, F)$. On a $J$-vacuum, $(F-J)$ flavors become massive and the CS level gets shifted accordingly. In each vacuum the low energy theory is the product of an $\mathcal{N} = 1$ topological sector and a NLSM. In theory A we find

$$\mathcal{N} = 1 \quad Sp(k-J)_{N-F-\frac{k-J-3}{2}} \quad \times \quad \text{NLSM} \quad \frac{Sp(F)}{Sp(J) \times Sp(F-J)} \, . \tag{6.15}$$

As it was the case for the domain walls of $SU(N)$ SQCD, some vacua break supersymmetry and get lifted. Recalling that an $\mathcal{N} = 1 \ Sp(m)_h$ CS gauge theory breaks supersymmetry if $|2h| < m + 1$, we find that supersymmetric vacua correspond to $J \ge F + k - N - 1$. Therefore, the full set of vacua is parameterized by $J$ in the interval

$$\max(0, F + k - N - 1) \le J \le \min(k, F) \, . \tag{6.16}$$

Table 3: Domain walls of massive 4d $\mathcal{N} = 1$ $Sp(N)$ SQCD with $F < N + 1$ flavors. Behavior of the conjectured 3d dynamics for $m < 0$, as $k$ and $J$ are varied.

|  | HGr$(0,F)$ trivial | HGr$(1,F)$ $\leftrightarrow$ HGr$(F-1,F)$ | HGr$(2,F)$ $\leftrightarrow$ HGr$(F-2,F)$ | $\cdots$ | HGr$(F-1,F)$ $\leftrightarrow$ HGr$(1,F)$ | HGr$(F,F)$ trivial |
|---|---|---|---|---|---|---|
| trivial $Sp(N-F+1)_{\frac{F-N-2}{2}}$ |  | $k=1$ | $k=2$ | $\cdots$ | $k=F-1$ | $k=F$ |
| $Sp(1)_{N-F+1}$ $\leftrightarrow$ $Sp(N-F)_{\frac{F-N-3}{2}}$ | $k=1$ | $k=2$ | $k=3$ | $\cdots$ | $k=F$ | $k=F+1$ |
| $Sp(2)_{N-F+\frac{1}{2}}$ $\leftrightarrow$ $Sp(N-F-1)_{\frac{F-N-4}{2}}$ | $k=2$ | $k=3$ | $k=4$ | $\cdots$ | $k=F+1$ | $k=F+2$ |
| $\vdots$ | $\vdots$ | $\vdots$ | $\vdots$ | $\ddots$ | $\vdots$ | $\vdots$ |
| $Sp(N-F)_{\frac{N-F+3}{2}}$ $\leftrightarrow$ $Sp(1)_{F-N-1}$ | $k=N-F$ | $k=N-F+1$ | $k=N-F+2$ | $\cdots$ | $k=N-1$ | $k=N$ |
| $Sp(N-F+1)_{\frac{N-F+2}{2}}$ trivial | $k=N-F+1$ | $k=N-F+2$ | $k=N-F+3$ | $\cdots$ | $k=N$ |  |
|  | $J=0$ | $J=1$ | $J=2$ | $\cdots$ | $J=F-1$ | $J=F$ |

This bound is the analog of (4.13) for $SU(N)$ SQCD.

Similarly, in theory B the low energy theories in supersymmetric vacua are

$$\mathcal{N} = 1 \quad Sp(N+1-k-H)_{F-1-\frac{N+k+H}{2}} \quad \times \quad \text{NLSM} \quad \frac{Sp(F)}{Sp(F-H) \times Sp(H)}, \qquad (6.17)$$

with $H$ in the interval

$$\max(0, F-k) \leq H \leq \min(N+1-k, F). \qquad (6.18)$$

It is easy to check—using the level/rank duality (6.12)—that these vacua exactly match with the supersymmetric vacua of theory A, upon the identification $H = F - J$.

We can collect into a table all vacua (6.15) that we found in the theories (6.9) at $m < 0$ as we vary $k$, with $0 < k < N + 1$. They describe all $k$-walls of $Sp(N)$ SQCD with $F < N + 1$ flavors, in the regime $m_{4d} \ll \Lambda$. Since the gauge factor in (6.15) only depends on $k - J$ while the NLSM only depends on $J$, we can set up a table, analogous to Table 1 for $SU(N)$ SQCD, where $J$ runs from 0 to $F$ horizontally, while $k - J$ runs from 0 to $N - F + 1$ vertically. The result is Table 3, which is $(N - F + 2) \times (F + 1)$, and it is the same as the table of domain walls of 4d $\mathcal{N} = 1$ $SU(N + 1)$ SQCD with $F$ flavors, provided one replaces the $U(n)$ gauge theories with $Sp(n)$ ones, and the Grassmannians with quaternionic Grassmannians.

Notice that taking $k = 0$ or $k = N+1$, the theory (6.9) has a single, trivially gapped vacuum at both $m \gtrless 0$ and there is no phase transition at $m = 0$. This corresponds to the fact that, formally, for $k = 0$ or $k = N + 1$ there is no domain wall at all. These two cases correspond to the two empty cells in Table 3.

Performing a similar analysis as it was done in Section 5 for $SU(N)$ SQCD, one can show that all vacua of the 3d theory (6.9) precisely match those obtained by solving the BPS domain wall equations of $Sp(N)$ SQCD in the small mass regime.

**Supersymmetry enhancement.** We conjecture that the theories (6.9) have enhanced $\mathcal{N} = 2$ supersymmetry at the CFT point for $F = 1$. The special case $N = F = 1$, corresponding to

the domain wall of 4d $SU(2)$ SQCD with 1 flavor, was already discussed around (4.21) and conjectured to have enhanced $\mathcal{N} = 4$ supersymmetry.

To understand the supersymmetry enhancement, we need to list the operators invariant under the symmetries that we can construct with $X$. There is only one quadratic operator invariant under $Sp(k) \times U(F)$:

$$\mathcal{O}_{(2)} \equiv X_{ai} X^*_{ai} \,. \tag{6.19}$$

It turns out that this is automatically invariant under $Sp(k) \times Sp(F)$. Indeed, using the extended notation, we have

$$\mathcal{O}_{(2)} = -\frac{1}{2} \operatorname{Tr} X^2 \Omega \,. \tag{6.20}$$

Next, there are three quartic operators that are invariant under $Sp(k) \times U(F)$:

$$\mathcal{O}^2_{(2)} = X_{ai} X^*_{ai} X_{bj} X^*_{bj} \,, \quad \mathcal{O}_{(4A)} \equiv X_{ai} X^*_{aj} X_{bj} X^*_{bi} \,, \quad \mathcal{O}_{(4B)} \equiv X_{ai} \Omega^{ab} X_{bj} X^*_{ci} \Omega^{cd} X^*_{dj} \,. \tag{6.21}$$

The first one is the "double trace" operator, and it preserves $Sp(k) \times Sp(F)$. One combination of the other two is the "single trace" operator that preserves $Sp(k) \times Sp(F)$:

$$\mathcal{O}_{(4A)} + \mathcal{O}_{(4B)} = \frac{1}{2} \operatorname{Tr} X^2 \Omega X^2 \Omega \,. \tag{6.22}$$

Both $\mathcal{O}_{(4A)}$ and $\mathcal{O}_{(4B)}$, taken separately, only preserve $Sp(k) \times U(F)$.

For small values of $k$ or $F$, some of these operators coincide. For $k = 1$, the combination $X_{aI} \Omega^{IJ} X_{bJ}$ is a $2 \times 2$ antisymmetric matrix and must be proportional to $\Omega^{ab}$. Indeed

$$X_{aI} \Omega^{IJ} X_{bJ} = \mathcal{O}_{(2)} \Omega^{ab} \,. \tag{6.23}$$

Substituting into the single-trace $Sp(F)$ invariant, we find the relation

$$\mathcal{O}^2_{(2)} = \mathcal{O}_{(4A)} + \mathcal{O}_{(4B)} \,. \tag{6.24}$$

Therefore, there are two quartic operators that preserve at least $Sp(1) \times U(F)$, but one linear combination preserves $Sp(1) \times Sp(F)$.

On the other hand, for $F = 1$, directly from (6.21) we see that

$$\mathcal{O}^2_{(2)} = \mathcal{O}_{(4A)} = X_a X^*_a X_b X^*_b \,, \qquad \mathcal{O}_{(4B)} = 0 \,. \tag{6.25}$$

Therefore, there is only one quartic operator invariant under at least $Sp(k) \times U(1)$, and it is automatically invariant also under $Sp(k) \times Sp(1)$. In other words, insisting on $Sp(k)$, quartic operators cannot break $Sp(F)$ to $U(F)$ when $F = 1$. Using again the extended notation we have

$$X_{aI} X_{bJ} \Omega^{ab} = \mathcal{O}_{(2)} \Omega^{IJ} \,, \tag{6.26}$$

which gives $\operatorname{Tr} X^2 \Omega X^2 \Omega = 2\mathcal{O}^2_{(2)}$, compatible with the relations above.

Now, consider a 3d $\mathcal{N} = 2$ $Sp(k)$ CS gauge theory with $F$ flavors $X$ in the fundamental representation. In the absence of holomorphic superpotential, the theory has $U(F)$ flavor symmetry, unless $F = 1$. Indeed, in $\mathcal{N} = 1$ notation there is a bare real superpotential

$$\mathcal{W}_{\mathcal{N}=1} = g_{\text{YM}} \sum_{i=1}^{F} X^\dagger_i \Psi X_i - \frac{g^2_{\text{YM}} h}{2} \operatorname{Tr} \Psi^2 \,, \tag{6.27}$$

where $h$ is the CS level. The real multiplet $\Psi$ is in the adjoint representation of $Sp(k)$:

$$\Psi = \Psi^\dagger = \Omega \Psi^{\mathsf{T}} \Omega \,. \tag{6.28}$$

We proceed with integrating $\Psi$ out, but we should be careful about the constraint (6.28): it implies the projection

$$\sum_{i=1}^{F} X_i^{\dagger} \Psi X_i = \operatorname{Tr} X^{\dagger} \Psi X = \frac{1}{2} \operatorname{Tr} \Psi \big( X X^{\dagger} + \Omega X^* X^{\mathsf{T}} \Omega \big). \tag{6.29}$$

Integrating $\Psi$ out we obtain

$$\mathcal{W}_{\mathcal{N}=1} = \frac{1}{8h} \operatorname{Tr} \big( X X^{\dagger} + \Omega X^* X^{\mathsf{T}} \Omega \big)^2 = \frac{1}{4h} \big( \mathcal{O}_{(4A)} - \mathcal{O}_{(4B)} \big). \tag{6.30}$$

We see that, indeed, the theory has $U(F)$ flavor symmetry. The only exception is the case with a single flavor, $F = 1$, because then $\mathcal{O}_{(4B)} = 0$: the flavor symmetry is enhanced to $Sp(1) \cong SU(2)$.

The $\mathcal{N} = 1$ theories (6.9) with $F = 1$ have a single quartic superpotential term and preserve $Sp(1)$ flavor symmetry. It is very plausible, and we conjecture, that the coefficient of that term flows in the IR to the $\mathcal{N} = 2$ point. The case of $Sp(k)$ CS gauge theory with 1 flavor was studied in [67] at large CS level, and it was shown that the $\mathcal{N} = 2$ point is indeed attractive.

For $F > 1$, the $\mathcal{N} = 1$ theories (6.9) have $Sp(F)$ global symmetry and thus the RG flow cannot reach the $\mathcal{N} = 2$ point, which has only $U(F)$ global symmetry. In other words, the RG flow does not generate the $Sp(F)$-breaking term $\mathcal{O}_{(4B)}$ which is instead present at the $\mathcal{N} = 2$ point. From this point of view, the theories with $k = 1$ are not special and do not enjoy supersymmetry enhancement.

## Acknowledgments

We are grateful to Bobby Acharya, Riccardo Argurio, Adi Armoni, Sergio Cecotti, Zohar Komargodski, and Itamar Shamir for enlightening conversations, suggestions or correspondence. Support by MIUR PRIN Contract 2015 MP2CX4 "Non-perturbative Aspects Of Gauge Theories And Strings" and by INFN Iniziative Specifiche ST&FI and GAST is acknowledged. F.B. and S.B. are also supported by the MIUR-SIR grant RBSI1471GJ "Quantum Field Theories at Strong Coupling: Exact Computations and Applications" and V.B. by the ERC-STG grant 637844-HBQFTNCER.

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
