# Peer review of "Living on the walls of super-QCD"

_SciPost Physics, doi:SciPost Phys. 6, 044 (2019)_

## Round 2 · Referee Report · Anonymous (Referee 1) · 2019-2-8

Strengths

see report

Weaknesses

see report

Report

The paper "Living on the walls of super-QCD", written by four authors whose surn
ame starts with B, concerns the 3d theory that lives on the walls of SQCD with F
<N. The theory admits an interesting structure.
For large and positive quark mass the 3d was found by Acharya-Vafa long time ago
. For small quark mass the authors find an interesting Grassmanian coupled to a
TFT. There is a phase transition in between.
The paper is very interesting and contains nice physics about the interplay betw
een 4d bulk physics and 3d worldvolume domain walls physics.
The paper is also nicely written.
I therefore recommend its publication.

Requested changes

n/a

---

## Round 2 · Referee Report · Anonymous (Referee 2) · 2019-2-18

Strengths

1) The paper contains an independent analysis from 3d and 4d perspectives that provides non-trivial cross-checks

2) It supplements recent results on 3d N=1 IR dualities with a natural 4d embedding

3) Clarity of presentation.

Weaknesses

It is not entirely clear if the 3d semi-classical analysis of vacua is complete.

Report

In this nicely written paper the authors study the low-energy dynamics on half-BPS domain walls in four-dimensional N=1 massive SQCD with gauge group SU(N) and F<N flavors. The Sp(N) case is also discussed. They propose that the low-energy domain wall theory is a specific 3d N=1 Chern-Simons matter theory, which is a natural generalisation of previous proposals for the domain wall theories on half-BPS domain walls in pure SYM theory. The authors show that their proposal passes a number of non-trivial checks. Most notably, it obeys N=1 infrared dualities precisely as expected from the 4d dynamics and reduces to the expected CS theory at large 4d flavor mass. The analysis of vacua for negative 3d flavor masses is more intricate. A detailed semi-classical analysis of BPS domain wall solutions in the massive 4d SQCD theory in the regime of small flavor mass provides compelling evidence that supports the vacuum structure obtained from the analysis of the proposed 3d theory on the domain walls.

The paper is written clearly with succinct review sections that help the presentation. I did not spot any worrying presentation issues. A single minor typo that I spotted is a missing \epsilon in the second equation in (2.12).

The only issue I would like to bring up on a technical level is the following. The proposed 3d CS-matter theory includes a classically massless scalar multiplet $\Phi$ in the adjoint representation. It is known that quantum effects make this multiplet massive and the authors use this fact to integrate out $\Phi$. Their 3d analysis is performed exclusively in the 3d N=1 CS-matter theory with the adjoint scalar multiplet integrated-out. It is also known that the mass of the multiplet $\Phi$ can be made parametrically small when the rank N of the bulk gauge group is large. Therefore, it is not very clear to me how strongly one can justify the semi-classical integration of $\Phi$. Could the full quantum dynamics of $\Phi$ affect significantly any of the aspects of the 3d analysis performed by the authors, e.g. the analysis of the vacua performed in section 4.1? Could there be potential effects that are missed by (4.3), which focuses on classically marginal and relevant couplings? In principal, higher order superpotential terms could be relevant if there are large anomalous dimensions. It is mentioned in the second sentence of page 13 that ''Higher order terms in W are expected to be irrelevant at the point m = 0''. However, some of the most intricate results are obtained for m non-vanishing. If there are potential alternatives to their analysis the authors should indicate them. If there are no alternatives and the authors believe that their arguments are robust it would be useful to mention relevant arguments explicitly.

The paper contains a number of important results with several non-trivial cross-checks. These results will be useful in future explorations of 3d non-supersymmetric QFTs and their connections to 4d physics. Besides the embedding of recently discussed 3d N=1 infrared dualities in 4d QFT the authors present interesting observations on supersymmetry enhancement and perform computations both in SU(N) and Sp(N) SQCD. For these reasons I believe the paper deserves to be published, but before my final recommendation I would like the authors to address the question I raised above.

Requested changes

I would like the authors to address the question I raised in my report.

---

## Round 3 · Referee Report · Anonymous · 2019-3-24

Strengths

1) The paper contains an independent analysis from 3d and 4d perspectives that provides non-trivial cross-checks

2) It supplements recent results on 3d N=1 IR dualities with a natural 4d embedding

3) Clarity of presentation

Report

I would like to thank the authors for the detailed explanations. They provide a satisfactory response to my question.

As I noted in my first review, this is a very good paper with several interesting results and non-trivial cross-checks. It makes progress in an exciting subject. I am happy to recommend this paper for publication in its current version.

---

## Round 3 · Author Response

We thank the reviewers for their close reading of the paper and useful remarks. We edited the paper to address the concerns of the second reviewer. For the benefit of the reader, we include here our response to the second reviewer's comments.

  • Let us now comment on the weakness highlighted by the referee. The 3d low-energy theory we propose to describe the domain walls has a quartic superpotential interaction and is not UV complete. It should be thought of as an effective theory, for values of the mass close to the phase transition (notice that, in any case, the domain wall can have a 3d description only below some scale, set by the SQCD scale and the 4d quark mass). A possible UV completion is in terms of a similar theory with no quartic superpotential interactions, and with a bare mass close to the value corresponding to the phase transition. The analysis of the vacuum structure of this theory has been done by Choi et al. [JHEP 1810 (2018) 105], taking into account perturbative effects at large values of the fields. Their results agree with the semiclassical analysis we did using the effective description. We think this gives strong support to the claim that the analysis of vacua is complete.

  • Regarding the question raised by the reviewer, it would surely be interesting to perform a more complete analysis that includes the adjoint multiplet $\Phi$. In particular, one may worry that the theory with the adjoint has more vacua (in which $\Phi$ gets a VEV) than the low-energy theory with the adjoint integrated out. We do not have a rigorous answer, but we believe that various arguments support the belief that no other vacua exist.

1) Close to the phase transition and in the vacuum that hosts the CFT, our analysis is complete because, no matter how small is the mass of the adjoint, we can always integrate out the adjoint as long as we look at lower energy scales. This will be true even if we move a little bit away from the CFT by a mass deformation, as long as this is smaller than the adjoint mass.

2) If we make the mass of the 3d quarks larger in our effective description, we cannot really use the effective description anymore. In particular, we could use the UV complete theory with no quartic superpotential interactions and with the adjoint field -- whose bare mass is set to zero. For large positive values of the quark bare (i.e. UV) mass, the 3d quarks can be integrated out, leaving a theory with the adjoint. This theory has been carefully analyzed by Bashmakov et al. [JHEP 1807 (2018) 123], with the conclusion that, for zero bare adjoint mass, no other vacua exist.

3) It does not make sense to take the bare mass of the 3d quarks to be large and negative in the UV complete description, because such a regime does not pertain to the physics of the domain walls. In particular, when that mass (in the description with no quartic superpotential) is zero, some vacua run to infinity and the Witten index jumps. However, we know from 4d arguments that the Witten index on the domain walls cannot jump. Indeed, probably that mass vanishes when the 4d mass vanishes, but in this case there are no 4d vacua nor domain walls, and so that value cannot really be reached.

4) Therefore, the regime in which the bare mass in the UV complete description is between zero and the positive value corresponding to the phase transition, is the one where in principle there could be other vacua once the adjoint is taken into account. We agree that a more complete analysis would be required to settle the issue. However, notice that the potential extra vacua should contribute a total of zero to the Witten index, because the analysis for large positive values of the mass was reliable, and the corresponding Witten index is already saturated by the vacua we found. This observation makes the existence of extra vacua unlikely (in our opinion), but of course not impossible.

We hope this addresses the reviewer’s concern.

---

## Round 3 · List of Changes

- Eq. (2.12) had indeed a missing \epsilon. Thanks for spotting it out. We corrected the equation.

- We have changed the first line after eqn. (5.1) for the sake of clarity.

- We have corrected a typo in the second line of page 26: W(Phi) -> W(\tilde M)

- We have added a footnote (now number 10) at page 12 to take into account the second reviewer's comment.

You are currently on this page

Resubmission 1812.04645v3 on 19 March 2019

---

## Editorial Decision

published